

# Radiative properties of mid-latitude cirrus clouds derived by automatic evaluation of lidar measurements

Erika Kienast-Sjögren[1,2], Christian Rolf[3], Patric Seifert[4], Ulrich K. Krieger[1], Bei P. Luo[1,5], Martina Krämer[3], and Thomas Peter[1]

[1]Institute for Atmospheric and Climate Science, ETH Zurich, Switzerland
[2]Now at: Fed. Office of Meteorology and Climatology, MeteoSwiss, Zurich Airport, Operation Center 1, CH-8058 Zurich, Switzerland
[3]Institute for Energy and Climate Research, Stratosphere, Forschungszentrum Jülich, Jülich, Germany
[4]Institute for Tropospheric Research (TROPOS), Leipzig, Germany
[5]Physical Meteorological Observatory Davos, PMOD WRC, CH-7260 Davos, Switzerland

*Correspondence to:* Erika Kienast-Sjögren (Erika.Kienast@meteoswiss.ch)

**Abstract.** Cirrus, i.e. high thin clouds that are fully glaciated, play an important role in the Earth's radiation budget as they interact with both long- and shortwave radiation and determine the water vapor budget of the upper troposphere and stratosphere. Here, we present a climatology of mid-latitude cirrus clouds measured with the same type of ground-based lidar at three mid-latitude research stations: at the Swiss high alpine Jungfraujoch station (3580 m a.s.l.), in Zürich (Switzerland, 510 m a.s.l.) and in Jülich (Germany, 100 m a.s.l.). The analysis is based on 13'000 hours of measurements from 2010 - 2014. To automatically evaluate this extensive data set, we have developed the "Fast LIdar Cirrus Algorithm" (FLICA), which combines a pixel-based cloud-detection scheme with the classic lidar evaluation techniques. We find mean cirrus optical depths of 0.12 on Jungfraujoch and of 0.14 and 0.17 in Zürich and Jülich, respectively.

Above Jungfraujoch, subvisible cirrus clouds ($\tau < 0.03$) have been observed during 7% of the observation time, whereas above Zürich and Jülich significantly less. From Jungfraujoch, clouds with $\tau < 10^{-3}$ can be observed three times more often than over Zürich and Jülich, and clouds with $\tau < 2 \times 10^{-4}$ even ten times more often. Above Jungfraujoch, cirrus have been observed to altitudes of 14.4 km a.s.l., whereas only to about 1 km lower at the other stations. These features highlight the advantage of the high-altitude station Jungfraujoch, which is often in the free troposphere above the polluted boundary layer, thus allowing to perform lidar measurements of thinner and higher clouds. In addition, the measurements suggest a change in cloud morphology at Jungfraujoch above $\sim$13 km, possibly because high particle number densities form in the observed cirrus clouds, when many ice crystals nucleate in the high supersaturations following rapid uplifts in lee waves above mountainous terrain.

The retrieved optical properties are used as input for a radiative transfer model to estimate the net cloud radiative forcing, CRF$_{NET}$, for the analysed cirrus clouds. All cirrus detected here have a positive CRF$_{NET}$. This confirms that these thin, high cirrus have a warming effect on the Earth's climate, whereas cooling clouds typically have lower cloud edges too low in altitude





to satisfy the FLICA criterion of temperatures below -38° C. We find $CRF_{NET} = 0.9$ Wm$^{-2}$ for Jungfraujoch and 1.0 Wm$^{-2}$ (1.7 Wm$^{-2}$) for Zürich (Jülich). Further, we calculate that subvisibe cirrus ($\tau < 0.03$) contribute about 5%, thin cirrus ($0.03 < \tau < 0.3$) about 45% and opaque cirrus ($0.3 < \tau$) about 50% of the total cirrus radiative forcing.

# 1 Introduction

One of the large challenges in climate modeling, characterized by a low level of scientific understanding, are clouds and their effects on climate (Dessler and Yang, 2003; Solomon et al., 2007; Boucher et al., 2013). This concerns also the microphysical processes leading to cirrus formation. These processes are subject to uncertainties in the understanding and parametrization of homogeneous and heterogeneous nucleation (e.g., Cirisan et al., 2014). For any specific cloud scene, unless there are in situ measurements, there is either no or incomplete knowledge of the number of ice nuclei (IN), the intensity of small-scale temperature fluctuations or the corresponding accurate values of upper tropospheric humidity (e.g., Ickes et al., 2015; Kienast-Sjögren et al., 2015).

Cloud properties such as cloud particle number, size and ice particle shape determine ice water content and optical depth, which together with the temperature of the cirrus cloud top determines whether the net cloud radiative forcing, $CRF_{NET}$, is positive or negative, i.e. whether a particular cirrus cloud is warming or cooling (Platt and Harshvardhan, 1988; Ebert and Curry, 1992; Lin et al., 1998a; Chen et al., 2000; Corti and Peter, 2009). The fact that liquid clouds contain spherical particles helps estimating their microphysical and radiative properties. Conversely, the different shapes and orientations (Pruppacher and Klett, 1997) of ice particles affect the extinction of light, complicating the estimation of the cirrus climate effect (Fu and Liou, 1993; Liou, 2002). Previous studies of the radiative effect of cirrus (e.g., Chen et al., 2000; Fusina et al., 2007; Cziczo and Froyd, 2014) have identified a range of several watts per square meter (Wm$^{-2}$) depending on the ice crystal number in a cirrus as compared to having an ice-free supersaturated region.

Lidar (LIght Detection And Ranging) measurements can be used to establish long time series of aerosol or cloud measurements. From the co- and cross-polarized components of the backscattered light the profile of the depolarization ratio can be obtained providing information about the sphericity of the retrieved particles, and thus on their liquid or solid state. Several lidar stations have applied their measurements of elastically backscattered light to investigate the properties of mid-latitude cirrus clouds. See Table 1 for an overview.

Here we present a cirrus cloud climatology based on 13'000 hours of lidar measurements from three mid-latitude sites; Jungfraujoch, Zürich and Jülich. The lidar technique is briefly described in Section 2.1. In Section 2.2, the newly developed evaluation algorithm FLICA is presented. Using FLICA we are able to analyze extensive lidar measurements automatically. The climatology of this data is presented in Section 3. We then apply the radiative transfer model of Corti and Peter (2009) to estimate the cloud radiative forcing caused by the detected cirrus clouds in Section 4. The results are compared to previous



**Table 1.** Lidar stations that have been used for systematic climatological studies of cirrus clouds in the mid-latitudes

| Measurement site | Location | Altitude [m a.s.l.] | Observation Period | Wavelength [nm] | Hours of data | References |
|---|---|---|---|---|---|---|
| Salt Lake City USA | 42°N, 68°W | 1726 | 1986-1996 | 694 | 2200 | Sassen and Campbell (2001); Sassen and Benson (2001); Sassen and Comstock (2001); Sassen et al. (2003, 2007) |
| Punta Arenas Chile | 53°S, 71°W | 126 | 03-04 2000 | 355, 532 | 71 | Immler and Schrems (2002) |
| Prestwick Scotland | 56°N, 5°W | 7 | 9-10 2000 | 355, 532 | 74 | Immler and Schrems (2002) |
| Haute Provence France | 44°N, 6°E | 679 | 1997-2012 | 532,1064 | ∼7000 | Goldfarb et al. (2001); Hoareau et al. (2013) |
| Rome Tor Vergata Italy | 42°N, 13°E | 107 | 2007-2010 | 532 | 500 | Dionisi et al. (2013) |
| Clermont-Ferrand France | 46°N, 3°E | 420 | 2008-2014 | 355 | ∼2000 | Fréville et al. (2015) |
| Seoul South Korea | 37°N, 127°E | 116 | 2006-2009 | 532, 1064 | ∼1000 | Kim et al. (2014) |
| Jülich Germany | 51°N, 6°E | 95 | 2011-2013 | 355 | 3274 | This work, also Rolf (2012) |
| Zürich Switzerland | 47°N, 9°E | 509 | 2010-2013 | 355 | 4678 | This work |
| Jungfraujoch Switzerland | 47°N, 8°E | 3580 | 2010-2014 | 355 | 5170 | This work |



studies in Subsection 4.2. The influence of the thinnest, subvisible, cirrus clouds on the cirrus radiative forcing is examined in Subsection 4.3. Finally, the main findings are summarized in Section 5.

## 2 Lidar

### 2.1 Lidar technique

This work uses the commercially available elastic backscatter lidar Leosphere ALS 450. This lidar emits linearly polarized laser pulses with an energy of 16 mJ at a wavelength of 355 nm and a repetition rate of 20 Hz. The full-angle field of view of the receiver telescope and the laser beam divergence are 1.5 mrad and 0.3 mrad, respectively.

The Nd:YAG laser of the ALS 450 is powered by a flash lamp. The flash lamp has lifetime corresponding to $5 \times 10^7$ shots or
694 hours or a month of continuous operation. In order to save flash lamp lifetime, the ALS 450 operated at Zürich and on the Jungfraujoch was coupled to a Vaisala Ceilometer CL31, which is a simple, low-maintenance elastic backscatter lidar (with a pulse energy about three orders of magnitude lower than the ALS 450). We use the ceilometer to detect thick clouds at low altitudes. Once thick clouds are present at an altitude lower than 1 km above the station, the lidar is automatically switched off (this is the case at roughly 30-40% of the time), and it is automatically switched back on once the low-level clouds are gone. In
Jülich, where no ceilometer was available, the ALS 450 was operated manually and switched off and on after visual inspection.

The range-corrected signal $r^2 P(r)$ detected by the ALS 450 can be described with the lidar equation (Kovalev and Eichinger, 2004; Wandinger, 2005):

$$r^2 P(r) = C \times O(r)[\beta_m(r) + \beta_p(r)] \exp\left(-2\int_{r_0}^{r}[\alpha_m(r') + \alpha_p(r')]\,dr'\right). \tag{1}$$

where $\beta_m$ and $\beta_p$ describe the backscatter from molecules and particles and $\alpha_m$ and $\alpha_p$ specify molecular and particulate extinctions, i.e. light attenuation by scattering and absorption, and take changes of scatterer density with altitude into account. Instrumental properties are described by the constant $C$. $O(r)$ is the overlap function which describes the overlap between the laser footprint and the telescope field of view. For the ALS 450 the complete overlap is achieved at a distance of 450 m from the lidar. As we analyze cirrus clouds that occurred entirely at greater heights above the lidar, we do not need to consider the
overlap function.

The Leosphere ALS 450 measures the co- and cross-polarized components of the return signal. In order to solve equation 1 it is required to obtain the total signal from these two components. We calculate the total signal based on both channels as described by Rolf (2012).

From the detected co- and cross-polarized signal components the depolarization ratio can be obtained (Schotland et al., 1971).





Light that is scattered back by non-spherical particles changes its polarization state, whereas spherical particles do not change the state of polarization of the returned light. Therefore, the depolarization ratio provides information about the sphericity of the detected particles (Schotland et al., 1971; Kovalev and Eichinger, 2004). The cross-polarized signal from aspherical ice particles in thin cirrus often provides the better contrast than the parallel signal, a property we will use in our cloud retrieval

algorithm. The lidar is pointed to 5° off-zenith to avoid the effect of specular reflections of horizontally oriented ice crystal plates on the measured backscatter signal and depolarization ratio (Platt et al., 1978; Westbrook et al., 2010).

For our cloud detection scheme elaborated in Subsection 2.2, we use the backscatter ratio (BSR) defined as:

$$\text{BSR} = \frac{\beta_p + \beta_m}{\beta_m}. \qquad (2)$$

To solve the lidar equation (Eq. 1) with four unknowns ($\beta_m$, $\beta_p$, $\alpha_m$ and $\alpha_p$) and only one measurement $r^2 P(r)$, we need to make use of best current knowledge. The molecular quantities $\beta_m$ and $\alpha_m$ are calculated from analysis data of the numerical weather prediction model (NWP) COSMO-2. We use pressure ($p$) and temperature ($T$) from COSMO-2 (COSMO, 2015) to calculate the molecular density of air and determine $\beta_m$ and $\alpha_m$ using Rayleigh theory (Bucholtz, 1995).

For the solution of Eq. (1) we use a lidar retrieval as described in Kovalev and Eichinger (2004). To ensure stable solutions, we use a far end boundary condition (Klett, 1981). Further, we need to define the extinction-to-backscatter ratio (hereafter referred to as lidar ratio). We derive $\epsilon = 0.234$, the anisotropy of the molecules present in the atmosphere, from Eq. (6) in She (2001) and Table 1 in Bucholtz (1995) for our lidar wavelength of 355 nm. The lidar ratio of the molecular part is evaluated as:

$$L_m = \frac{8\pi}{3} \times \frac{180 + 40\epsilon}{180 + 7\epsilon} \approx 8.7, \qquad (3)$$

where $\sigma^R$, given by Eq. (6) of She (2001), is divided by the expression for $\sigma_\pi^C$, provided in Eq. (4) of She (2001), as the receiver optical bandpass spectral width of 0.3 nm ($<24 \text{ cm}^{-1}$ at $28170 \text{ cm}^{-1}$) suppresses the rotational Raman wing spectral contribution (Arshinov and Bobrovnikov, 1999). Note that our $\epsilon$ is called $R_A$ by She (2001). The particulate lidar ratio is defined as:

$$L_p = \frac{\alpha_p}{\beta_p} \qquad (4)$$

Several studies have been performed to measure the particulate lidar ratio of cirrus clouds (e.g., Ackermann, 1998; Immler and Schrems, 2002; Larchevêque et al., 2002; Seifert et al., 2007). It can be obtained directly from Raman lidars that allow for an independent measurement of particle extinction and backscatter coefficients (Cooney, 1972; Giannakaki et al., 2007; Radlach et al., 2008; Reichardt et al., 2002; Achtert et al., 2013). In our retrieval we determine the lidar ratio such that BSR=1 above and below the cirrus cloud (e.g., Rolf, 2012).

The lidar equation (Eq. 1) assumes single-scattering of the emitted light in the direction $180°$ to the emitted direction only. In reality, this is not strictly the case. As seen in Fig. 1 in Wandinger (1998), cloud particles produce strong forward scattering.





This causes some of the scattered photons to remain within the field of view of the lidar, where they can be scattered back to the lidar receiver during a subsequent scattering process. These additional backscattered photons cause an underestimation of the particle extinction. The strength of multiple scattering depends mainly on the laser divergence, the telescope field of view and the effective radius of the scattering particles. In order to provide extinction values that are comparable to other lidar systems

and cloud conditions, the measured apparent, multiple-scattering affected extinction coefficient $\alpha_{\mathrm{p}}^{\mathrm{obs}}$ needs to be corrected with the correction factor $\gamma$ to obtain single-scattering related values $\alpha_{\mathrm{p}}^{\mathrm{single}}$, such that

$$\alpha_{\mathrm{p}}^{\mathrm{single}} = \frac{\alpha_{\mathrm{p}}^{\mathrm{obs}}}{\gamma}. \tag{5}$$

We use the multiple scattering model by Hogan (2008) as described by Wandinger (1998) and Seifert et al. (2007) to derive $\gamma$. The effective radius of the cirrus particles is taken from a climatology provided by Wang and Sassen (2002). For particles

much larger than the detection wavelength, as it is the case for ice crystals observed with lidar, about 50% of the scattering occurs into the forward direction. In this study we find an average value for $\gamma$ of 0.56 for Jungfraujoch and 0.52 (0.54) for Zürich (Jülich).

The lidar retrieval poses several uncertainties. Using NWP-data to calculate the molecular properties results in a maximal

error of 2 %. However, there are uncertainties pertained to the data themselves. The lidar detector counts photons, and we calculate the counting error by means of poisson statistics. The assumed lidar ratio is also an error source. Here, we use lidar ratios that deviate $\pm 5$ sr from the determined lidar ratios to assess for the uncertainty caused by determining a lidar ratio. To assess the total, maximum uncertainty, we sum up the individual contributions to the uncertainty. Seifert et al. (2007) estimated the error in the multiple-scattering correction in the order of 10%. The signal-to-noise ratio (SNR) is determined by the variation

within the 5-min-average profiles. As the Leosphere lidar does not allow to retrieve the photon counts directly from the data, we calculate the SNR from the original lidar profiles as

$$\mathrm{SNR} = \sqrt{N} \cdot \frac{\mathrm{mean}(r^2 P(r))}{\mathrm{std}(r^2 P(r))}, \tag{6}$$

where $\mathrm{mean}(r^2 P(r))$ is the mean range-corrected signal and $\mathrm{std}(r^2 P(r))$ the standard deviation of the originally retrieved lidar profiles over $N = 600$ shots (following a suggestion by P. Royer, Leosphere, private communication 11.8.2014).

## 2.2   Cirrus Detection Algorithm FLICA

For an efficient evaluation of this extensive data set, the automated data evaluation algorithm FLICA (Fast LIdar Cirrus Algorithm) was developed. The algorithm is based on a classical lidar retrieval (e.g., Klett, 1981; Kovalev and Eichinger, 2004)

combined with a cloud detection scheme. FLICA analyzes profiles over 5 minutes (i.e. averages over 5 min $\times$ 60 s/min $\times$ 20 shots/s = 6000 shots). This time range was chosen to be short enough to ensure that all clouds could be detected by the algorithm while long enough to provide profiles smooth enough for the lidar retrieval to function. The 5-min profiles of the





lidar measurements are further smoothed using a boxcar filter in the vertical coordinate over 150 m and 5 profiles in time to reduce the noise level and hence simplify the automatic evaluation by FLICA.

The output of the cloud detection scheme has been visually inspected for individual days and was found not to show any apparent artifacts. There is a trade-off between detecting cloud structures small enough and avoiding misclassifying noise as a cloud, especially for daytime measurements. The combination of the criteria below represents a rather conservative approach, which might result in missing some particularly small/thin clouds. The conservative approach ensures that no noise is misclassified as a cirrus cloud. An example of the resulting cloud detection can be seen in Fig. 1.

The FLICA algorithm contains the following steps:

I **Cloud top detection.** The cloud top is needed as an upper boundary for the subsequent lidar retrieval. Our cloud top detection averages individual lidar profiles so that the resolution of one pixel is 5 min in time and 30 m in altitude. Areas of $3 \times 3$ pixels are examined, with the pixel to be checked for cloudiness in the center. At least 8 of the 9 examined pixels have to have a volume depolarization larger than 0.007(0.006) for day(night)-time measurements. At least 8 of 9 pixels also have to have larger co- and cross-polarized raw signals than empirical thresholds.

II **Setting the far-end boundary condition for the lidar retrieval.** The mean of the co-polarized signal at altitudes from detected cloud top to 500 m above cloud top is computed for each profile individually and used as far-end boundary condition for the lidar retrieval as described in Klett (1981). At this boundary, we assume a BSR of 1. This assumption introduces no error if the aerosol density above the cloud is the same as the one of the interstitial aerosol. If these densities are different we estimate from our in-situ observations (http://www.iac.ethz.ch/groups/peter/research/Balloon_soundings/ COBALD_sensor) that the error introduced is of the order of 1 - 2 %.

III **Lidar retrieval.** The lidar retrieval is performed as described in Chapter 5 in Kovalev and Eichinger (2004) to solve the lidar equation (Eq. 1) and calculate the extinction coefficients and BSR of the cirrus cloud. The retrieval is performed for a set of lidar ratios (5:5:40) and the best choice was determined such that BSR is closest to 1 below the cirrus cloud. The BSR is corrected during the retrieval such that the mean BSR in the range 500 m above the cloud top is equal to 1.

IV **Cirrus cloud detection.** The cloud detection scheme is based on the retrieved BSR and volume depolarization as follows:

Resolution of one pixel: 5 min in time, 30 m in altitude.

Areas of $3 \times 3$ pixels are examined, with the pixel to be checked for cloudiness in the center. At least 8 of the 9 examined pixels have to have a volume depolarization larger than 0.007(0.006) and a BSR larger than 1.08(1.03) for day(night)-time measurements.

Temperature has to be lower than -38° C to ensure pure ice clouds (this is checked using COSMO-2 or COSMO-7 analysis data).

The detection is applied to each pixel at each time independently.





Clouds extending less than 150 m in altitude during daytime conditions are not further taken into account (as noise-limiting measure), whereas nighttime clouds are allowed to be as thin as $3 \times 30$ m = 90 m.

Cloud pixels separated vertically by less than 150 m are merged into one cloud layer.

The detected cloud top and cloud base, $h_{top}$ and $h_{base}$, are stored.

VI   **Multiple scattering correction.** The single-scattering extinction coefficients are derived from the apparent, multiple-scattering affected extinction coefficients as described in Section 2.1. We use the multiple scattering model of Hogan (2008) as described in Wandinger (1998) and Seifert et al. (2007).

VII   **Optical depth.** The optical depth $\tau$ of the detected cirrus cloud is calculated by integrating over the retrieved extinction profiles.

$$\tau = \int\limits_{h_{base}}^{h_{top}} \alpha_p(r)dr \tag{7}$$

VIII   **Radiative effect.** The optical depth $\tau$ combined with temperatures from COSMO-2 or COSMO-7 are used to calculate the radiative effect of the cirrus cloud by means of the model of Corti and Peter (2009).

## 3   Lidar cirrus climatology

### 3.1   Measurement sites

Here we present the retrieved lidar cirrus climatology. First, a description of the different measurement sites shown in Fig. 2 is provided. Subsequently, we present the climatology of the cirrus cloud properties. The section ends with a comparison of our data with previous mid-latitude climatology studies.

### 3.1.1   Jungfraujoch

Jungfraujoch is the highest measurement site used in this study. It is located in the Swiss Alps (46.55° N, 7.99° E) at 3580 m
a.s.l. at the top of the Aletsch glacier. Due to its high elevation, the research station is frequently situated in the free troposphere (Zieger et al., 2012), which is a great advantage for lidar measurements. The sphinx observatory, where our measurements took place, is one of the Global Atmospheric Watch (GAW) research stations. Therefore, long time series of meteorological measurement data are available for this site. Due to its high location and cold climate, the site poses challenges for instruments being able to run continuously. The Leosphere ALS 450 used in this study was built into a ventilated, temperature-controlled
and regulated containment.





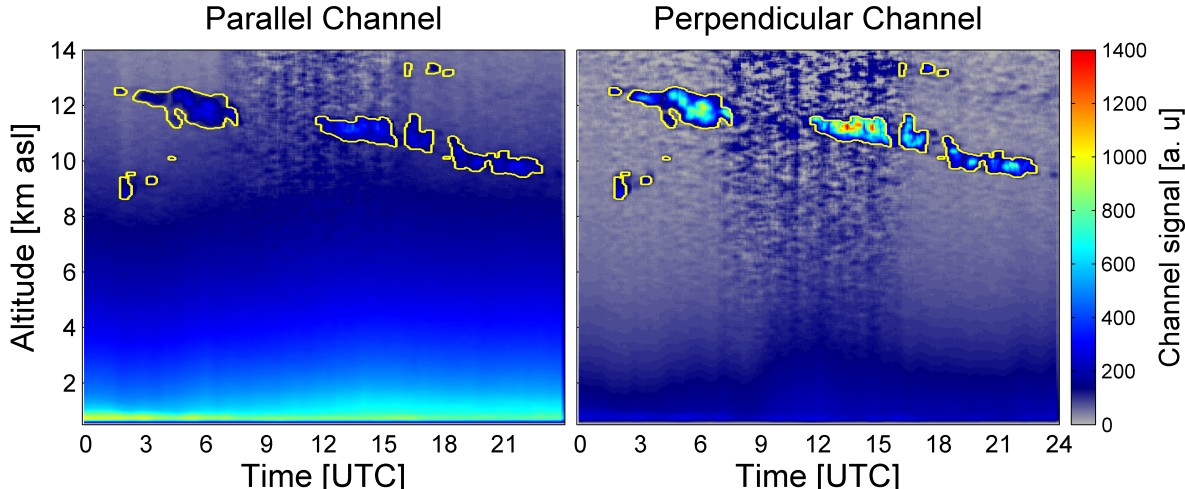





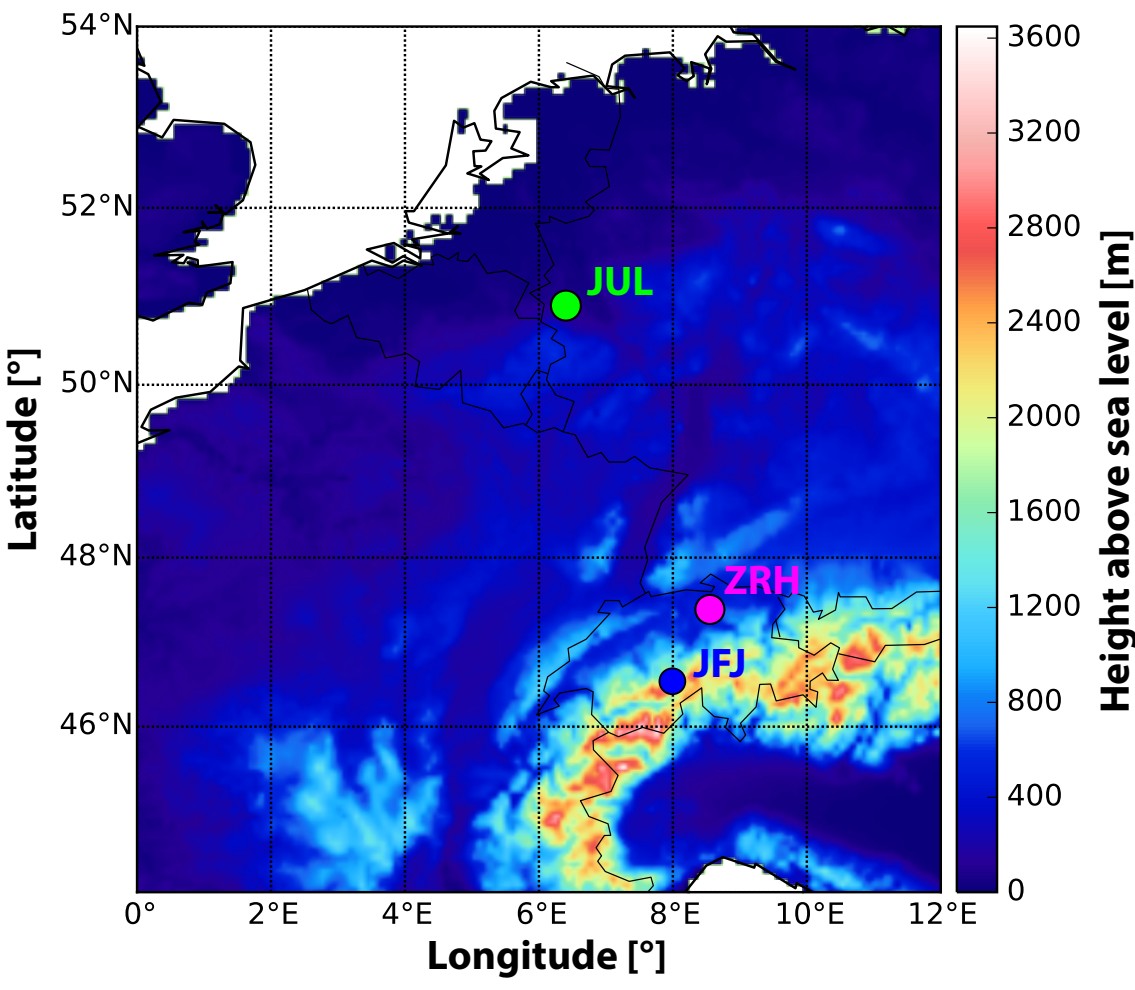

**Figure 2.** Location of the measurements sites Jülich(JUL), Jungfraujoch(JFJ) and Zürich(ZRH). Color-coded: Topography in COSMO-7. Black lines: National borders. Source: National Geophysical Data Center, 1993. 5-minute Gridded Global Relief Data (ETOPO5). National Geophysical Data Center, NOAA. doi:10.7289/V5D798BF [13.08.2015]



### 3.1.2 Zürich

Zürich (47.37° N, 8.55° E), the largest city in Switzerland, is situated in the northern part of Switzerland at 408 m a.s.l., within the Swiss Plateau. The Swiss Plateau is surrounded by the Alps and Jura mountains, which create a basin through which air masses originating from the Atlantic Ocean are funneled. Therefore, the predominant wind direction in Zürich is from the
southwest. Although the Swiss plateau is a large basin, it is still hilly. Lake Zürich is a basin itself within the Swiss plateau, and the city of Zürich is situated on the lake's northern shore. Lidar measurements were taken from the roof of ETH's Institute for Atmospheric and Climate Science (IAC), which is 500 m above sea level and located in the middle of Zürich. Aerosol particles in and around Zürich arise from industry, transportation and housing, and the large airport nearby. In contrast to Jungfraujoch, such additional aerosol sources cause low level extinction of the emitted laser pulse.

### 10 3.1.3 Jülich

The Research Center Jülich is located 91 m a.s.l. in the western part of middle Germany (50.91° N, 6.40° E) between the larger cities Aachen and Köln in North Rhine-Westphalia. Due to its low elevation and location close to the Netherlands, the weather fronts arrive more or less directly from the Atlantic Ocean without moderation by orography. The terrain around Jülich is relatively flat. The Research Center itself is located in a rural area and therefore the lidar measurements might be less
influenced by boundary layer aerosol than the Zürich measurements, despite nearby brown coal industry activity.

### 3.2 Cirrus Climatology

Following Section 2.2 we present the climatological evaluation of more than 13'000 hours of lidar measurements within the period 2010-2014 from the three mid-latitude measurement sites. The main information on the measurement statistics for the three sites is compiled in Table 2. More measurements are available from Jungfraujoch and Zürich than from Jülich. For
Jungfraujoch, most of the data was retrieved in the spring, whereas during the summer only very limited data is available. In Zürich, on the other hand, a large number of the measurements took place during the summer, while the other seasons show similar coverage. The Jülich lidar was running predominantly during spring and summer, while the autumn and winter data is sparse. The amount of data has to be considered when judging seasonal variability. The retrieved cirrus properties listed in Table 2, indicate a temporal cirrus cloud coverage between 9 and 15 % for all stations, agreeing well with the CALIPSO
measurements discussed by Sassen et al. (2008).



**Table 2.** Properties of the cirrus clouds detected between 2010-2014.

|  | JFJ | Zürich | Jülich |
|---|---|---|---|
| **General Properties** |  |  |  |
| Hours of measurements[1] | 5170 | 4678 | 3274 |
| Number of cirrus detected[2] | 10295 | 6021 | 7184 |
| Cirrus cloud coverage in %[3] | 14 | 9 | 15 |
| Low cloud coverage in %[4] | 15 | 8 | 26 |
| Clear sky in %[5] | 71 | 83 | 59 |
| **Fraction of measurement time by season in %** |  |  |  |
| DJF | 24 | 17 | 18 |
| MAM | 40 | 15 | 39 |
| JJA | 14 | 48 | 31 |
| SON | 22 | 20 | 12 |
| **Cloud occurrence frequencies in categories according to Sassen and Cho (1992) (expressed as fraction of "number of cirrus detected")** |  |  |  |
| Subvisible cirrus ($\tau < 0.03$) in % | 43 | 35 | 32 |
| Thin cirrus ($0.03 < \tau < 0.3$) in % | 46 | 52 | 51 |
| Opaque cirrus ($0.3 < \tau$) in % | 11 | 13 | 17 |
| Mean $\tau$[6] | $0.12^{+.02}_{-.06}$ | $0.14^{+.02}_{-.08}$ | $0.17^{+.02}_{-.08}$ |

[1] Refers to the number of hours lidar measurements with the ALS 450. Not included are times times when the ceilometer detected low-level clouds closer than 1.5 km.

[2] According to the specifications of the FLICA algorithm, see Subsection 2.2.

[3] This compares reasonably well with 11 % zonal average by Chen et al. (2000).

[4] Refers only to clouds at least 1 km above the lidar.

[5] As observed by the ALS 450.

[6] Uncertainties as described in the last paragraph of Subsection 2.1. Mean values compare reasonably well with monthly mean values of 10-20% from ISCCP (Soden and Donner (1994); also http://www.gfdl.noaa.gov/ice-clouds-in-the-skyhi-general-circulation-model).

The seasonal dependence of the observed cirrus coverage is displayed in Fig. 3. The most striking feature is the difference between the wintertime measurements in Zürich and Jungfraujoch showing a cirrus coverage of around 12%, while in Jülich

5 this is about 33%. This is in qualitative agreement with geographical maps of high cloud amount (cloud pressure smaller





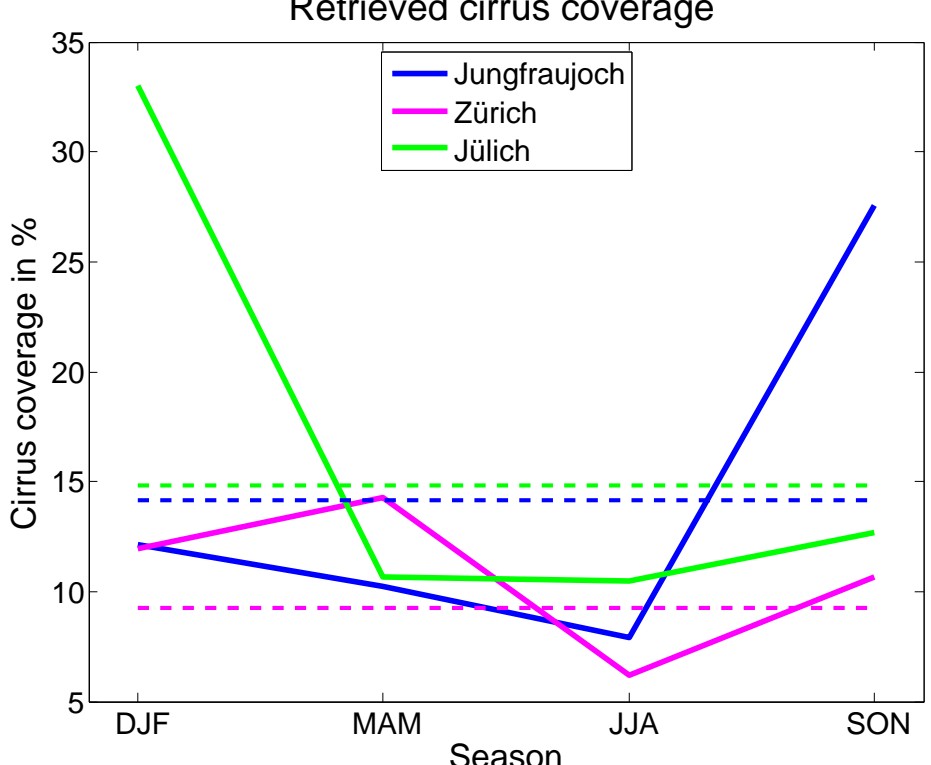

**Figure 3.** Seasonal cycle of cirrus cloud coverage for the three measurement sites. Dashed lines: annual means.

than 440 hPa) for January observed by the TIROS-N Operational Vertical Sounder, TOVS, averaged over 8 years, 1987-1995 (http://ara.abct.lmd.polytechnique .fr/index.php?page=clouds). For January, these data suggest decreasing amounts of high clouds when the air passes from the North Sea towards the Alps. Also, a time series of 40 years of measurements of turbidity in Jülich confirm the high cloud coverage during wintertime (personal communication A. Knaps, Forschungszentrum

5  Jülich). However, there are large uncertainties in this part of our climatology, as the number of hours of measurements with the Leosphere ALS 450 available for the Jülich winter is small (Table 2), the specific winter might have had a particularly high cirrus cloud coverage, and the applied manual operation of the lidar might add bias. Another remarkable feature is that autumn maximum in cirrus coverage observed above Jungfraujoch. Also this feature is in qualitative agreement with the seasonal cycle suggested by the TOVS data set, but again interannual variability might be important but cannot be derived from our measure-

10  ments.

The first property of interest is the distribution of the optical depths of the detected cirrus clouds, which we classify according to Sassen and Cho (1992) (cf. Table 2): clouds with an optical depth $\tau < 0.03$ are not visible to the naked eye, and





hence termed subvisible. Cirrus clouds with an optical depth $\tau$ in the range $0.03 \leq \tau < 0.3$ are termed thin, while clouds with $\tau \geq 0.3$ are referred to as opaque. The upper limit of detection for our lidars is $\tau \approx 3$, as for larger optical depths the light is almost fully extinguished within the cloud. Under these conditions no molecular signal from above the cloud can be detected (Immler and Schrems, 2002), as would be required for an inversion. Therefore, we are not able to specify the optical thickness

of the thickest cirrus clouds. Chen et al. (2000) classified clouds with tops above 440 hPa ($\approx 6500$ m) and optical depths larger than 3.6 as cirrostratus. These cirrus clouds may have a negative cloud radiative effect $\text{CRF}_{\text{NET}}$, but cannot be considered here because of the detection limits of our lidar instrument.

Figure 4b shows the optical depth of the retrieved cirrus clouds for different seasons. The grey dashed lines indicate the

categories defined by Sassen and Cho (1992). The optical depth averaged over the whole data sets for each measurement site is displayed in Figure 4a. The occurrence frequency of subvisible cirrus clouds is larger for Jungfraujoch than for the other two sites. Two reasons may be responsible for the observed differences. First, Jungfraujoch is located in the central Alps, where orography-driven lifting of air masses leads frequently to mountain-wave (lenticularis) cirrus. These clouds are thicker than large-scale cirrus clouds, but thinner than the cirrus formed as outflow of anvils or in warm conveyor belts. The second reason

is the enhanced detectability of optically thin clouds at Jungfraujoch as a result of improved signal-to-noise ratios (SNR, see Eq 6). The alpine site is located at an altitude of 3500 m asl, 3000 m above Zürich and 3400 m above Jülich. According to Eq. 1 the received signal depends on the inverse of the squared range between lidar and target. In addition, the Jungfraujoch is frequently above the boundary layer. Therefore, measurements from Jungfraujoch avoid strong beam extinction due to boundary layer aerosols.

Figure 5 provides vertical profiles of the cloud-mean signal-to-noise ratio (SNR) of the three stations, where the noise is obtained from Eq. 6. From the profiles it can be seen that the SNR of Jungfraujoch extends to greater heights by about 3 km. This suggests that the increased detection rate of thin and subvisible cirrus clouds is a result of the increased signal-to-noise ratio. Furthermore, the SNR at Jungfraujoch increases at heights above 13 km a.s.l. This suggests that the morphology of the

clouds at these heights differs from the morphology of the highest clouds observed at Jülich and Zürich. The backscattering efficiency appears to be enhanced in these clouds, possibly because a large amount of small crystals formed in the observed cirrus clouds, when many ice crystals nucleated in the high supersaturations in rapid uplifts as they occur in lee waves above mountainous terrain (Lin et al., 1998a, b; Kärcher, 2003).

The number of detected subvisible cirrus as function of optical depth and cloud top altitude is depicted in Fig. 6. As expected, Jungfraujoch displays a larger fraction of subvisible cirrus as well as higher cirrus cloud tops. Therefore, we have evidence for both:

    (a) the advantage of location of the higher Jungfraujoch as evidenced by the ability to measure thinner subvisible clouds (by about a factor 5);





**Figure 4.** Optical depths of the three measurement sites. (a): Means across whole data set. (b): Seasonal cycle of optical depth. Horizontal line in box: median. Boxes: the upper and lower quartile. Whisker: extremes. Gray horizontal lines: Cirrus categories by Sassen and Cho (1992).





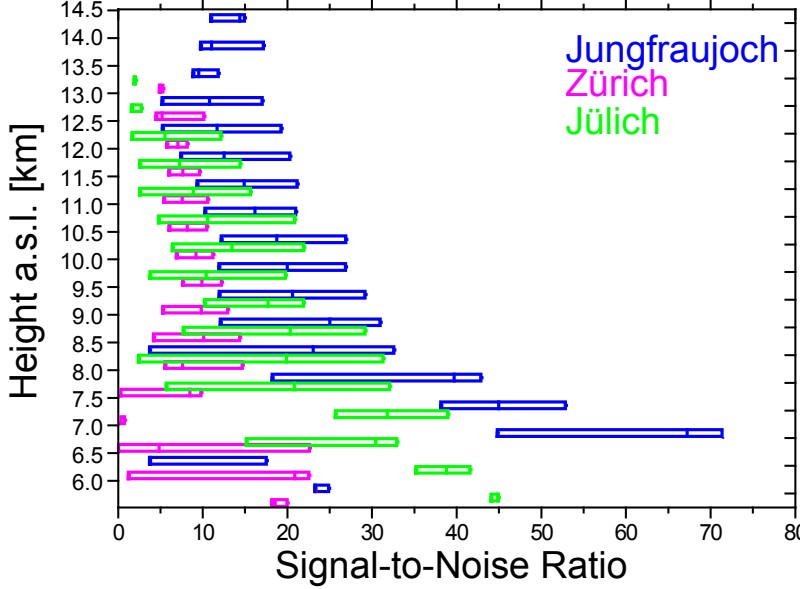

**Figure 5.** Vertical distribution of the signal-to-noise ratio of the detected cirrus clouds for Jungfraujoch, Jülich, and Zürich. The center line of each box plot represents the median. The left and right limits of the box plots mark the 25% and 75% percentiles, respectively.

    (b) the special conditions above Jungfraujoch caused by orographic forcing, which affects the morphology of the cirrus as evidenced by the enhanced SNR at high altitudes.

It is interesting to see that at Jungfraujoch the lower detection limit in optical depth of a few times $10^{-5}$ is approached in a few cases. However, by far the most subvisible cirrus stay clearly above $\tau = 10^{-4}$ proving that physical mechanisms prevent clouds

5   so thin to survive for appreciable times. Nucleation is one such mechanism. In case these clouds nucleate homogeneously, this is likely to happen in nucleation bursts, which will provide the newly formed clouds immediately a minimum optical depth. The same is true for heterogeneous nucleation, if the nucleation barrier and the number of nuclei are at all significant. One mechanism that might lead to extremely low ice crystal number densities is the formation of fall-streaks and subsequent dispersion of the particles. The rare occurrence of clouds with $\tau < 5 \times 10^{-4}$ suggests that such mechanisms do not often lead to

10   the formation of so thin clouds.

To ensure that the highest cirrus clouds observed above Jungfraujoch were not volcanic particles, we have examined satellite measurements and found no indication for volcanic influences. The effect of the high altitude of Jungfraujoch can also be seen in the cloud tops at the different measurement sites (see Fig. 7a). The cloud tops are higher above Jungfraujoch than above

15   Zürich and Jülich. The retrieved cloud tops agree well with the observations by Sassen and Campbell (2001) in Salt Lake City (40° N, 12° W, 1520 m a.s.l.) as well as by Hoareau et al. (2013) in Haute Provence ( 44° N, 6° E, 679 m a.s.l.).The data from Salt Lake City and from Haute Provence were evaluated using evaluation schemes differing from FLICA and differing



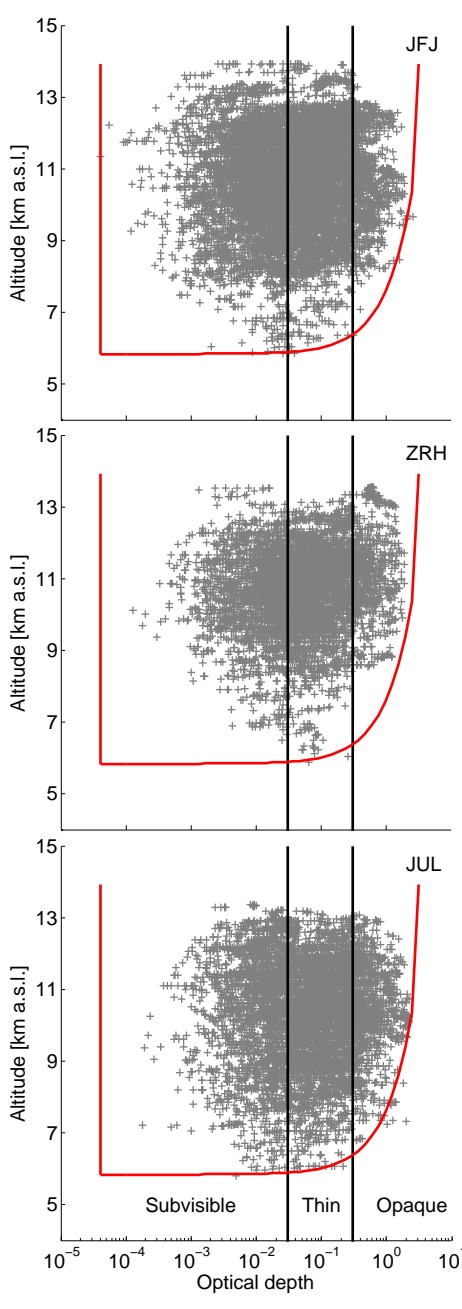

**Figure 6.** Scatter plots of cloud optical depths and cloud top altitudes for the cirrus detected above Jungfraujoch (JFJ), Zürich (ZRH) and Jülich (JUL). The red lines provide an indication of the range of data accessible by the lidar measurements: $AOD_{min} = 4 \times 10^{-5}$, $AOD_{max} = 2.6$, and $Alt_{min} = 5.8$ km. The lower edge of the accessible altitude is determined from $T < -38°C$. Thicker clouds are more likely to extend into lower, warmer levels and therefore are more likely to be excluded from the analysis.



**Figure 7.** (a) Cloud tops (in 500-m steps) at the three sites of this study as well as in Salt Lake City by Sassen and Campbell (2001) and Haute Provence by Hoareau et al. (2013). (b) Tropopause derived from COSMO regional weather forecast model analyses (2.2 km horizontal resolution for JFJ and ZRH, 6.6 km resolution for JUL).

amongst themselves, which may influence the results. However, the cloud top altitudes are very similar for the five mid-latitude stations. As Salt Lake City is located further south than the other sites, the slightly higher cloud tops may be a result of a higher tropopause being present over Salt Lake City compared to the other sites. Similarly, we see in Fig. 7b that the tropopause over Jülich, which is located further north than Zürich and Jungfraujoch, generally is lower. Between Zürich and Jungfraujoch, the

5  tropopause reaches similar altitudes with a larger spread over Jungfraujoch (possibly due to the Alpine heat low affecting the Jungfraujoch frequently during summer time).





## 4 Cirrus Radiative Forcing (CRF)

### 4.1 Method of Calculation

To quantify the net radiative effect, $CRF_{NET}$, for the cirrus clouds observed here we use the radiation model of Corti and Peter (2009), which is a simplified model based on the more sophisticated Fu-Liou radiative model (Fu and Liou, 1992, 1993). The

cloud radiative forcing due to shortwave radiation, $CRF_{SW}$, is dependent on the surface albedo, the solar zenith angle as well as the cirrus cloud optical depth $\tau$ (see Eq. 13 of Corti and Peter (2009) for details). The longwave cloud radiative forcing, $CRF_{LW}$, is mainly determined by the temperature difference between the Earth's surface and the cirrus cloud top temperature and by the cloud optical depth $\tau$ (see Eq. 6 of Corti and Peter (2009) for details). The net cloud radiative forcing, $CRF_{NET}$, is calculated as a superposition of these two effects (i.e. $CRF_{NET}=CRF_{SW}+CRF_{LW}$). The following parameters are needed as

input for the calculation of the radiative effect:

- Solar constant $S$

- Solar zenith angle $Z$

- The surface albedo $\alpha$

- The cloud optical depth $\tau$

- The surface temperature $T_{srf}$

- The cloud top temperature $T_{cld}$

The values of the solar constant $S$, multiplied by the fraction of the day that the sun is above the horizon, and the mean solar zenith angle $Z$ are set to 684 Wm$^{-2}$ and 60° (daily mean conditions with zero incoming flux at nighttime), respectively, as suggested by the online version of the radiation model (Corti and Peter, 2009b). This results in an incident solar flux $I = 684$

$\times\ 0.5 = 342$ Wm$^{-2}$. The amplitude of the solar background noise in the lidar signal profiles is used to distinguish between day and night time. We use an albedo of 0.3 (corresponding to the global average value). The cloud optical depth $\tau$ is automatically calculated in the FLICA for 5-min profiles as described in Section 2.2. The temperatures needed for the radiation model ($T_{srf}$ and $T_{cld}$) are extracted from the COSMO-2 (for Jungfraujoch and Zürich) and COSMO-7 (for Jülich) model. The radiation model of Corti and Peter (2009) is well suitable to be used with lidar data, as the model does not require further information,

such as ice crystal sizes or shapes, which the lidar measurements could not provide.

### 4.2 Comparison of CRFs with previous studies

We compare our computational results to satellite data, which have been averaged zonally at 50°N or globally and combined with a radiative transfer model (Chen et al., 2000). The results of this comparison are listed in Table 3 together with maximum possible uncertainty ranges (see last paragraph of Subsection 2.1). The "overcast values" (i.e. taking only cloudy values into

account) consider the radiative effect under conditions with cirrus clouds, while the "all sky values" include also conditions





**Table 3.** Cirrus radiative forcing at the Top of Atmosphere in $Wm^{-2}$ for Jungfraujoch, Zürich and Jülich, as compared to 50° N zonally averaged and globally averaged values provided by Chen et al. (2000). Small numbers in CRF-values indicate uncertainty ranges according to the last paragraph of Subsection 2.1. Small numbers in global cloud coverage indicates variability in zonal averages.

|  | JFJ | Zürich | Jülich | 50° N | global |
|---|---|---|---|---|---|
| Cirrus coverage in % | 14 | 9 | 15 | 11 | $13^{+7}_{-8}$ |
| Overcast CRF$_{NET}$ | $6.2^{+0.7}_{-3.0}$ | $10.6^{+1.5}_{-5.3}$ | $11.0^{+1.4}_{-4.9}$ | 2.0 | 5.4 |
| CRF$_{LW}$ | $7.2^{+1.0}_{-3.6}$ | $12.3^{+1.8}_{-6.1}$ | $13.3^{+1.6}_{-6.0}$ | 20.1 | 30.7 |
| CRF$_{SW}$ | $-1.0^{+0.5}_{-0.3}$ | $-1.7^{+0.8}_{-0.3}$ | $-2.4^{+1.1}_{-0.2}$ | -18.1 | -25.3 |
|  |  |  |  |  |  |
| All sky CRF$_{NET}$ | $0.9^{+0.1}_{-0.4}$ | $1.0^{+0.1}_{-0.5}$ | $1.6^{+0.2}_{-0.7}$ | 0.5 | 1.3 |
| CRF$_{LW}$ | $1.0^{+0.1}_{-0.5}$ | $1.1^{+0.2}_{-0.6}$ | $2.0^{+0.2}_{-0.9}$ | 3.0 | 5.5 |
| CRF$_{SW}$ | $-0.1^{+0.1}_{-0.0}$ | $-0.2^{+0.1}_{-0.0}$ | $-0.3^{+0.2}_{-0.0}$ | -2.5 | -4.2 |

without cirrus by considering the cirrus occurrence frequency. While the cirrus cloud coverage at 50°N from the satellite-based climatology ISCCP (Chen et al., 2000) is similar to our observations, the ISCCP category of cirrus clouds show much larger cloud radiative forcings in the short- and longwave, CRF$_{SW}$ and CRF$_{LW}$ (by more than one order of magnitude). This can only be explained in terms of a much larger optical depth $\tau$ of the clouds observed by the satellites.

The reason for this at first sight surprising differences are the different definitions of "cirrus". First, FLICA detects only clouds with lower cloud edge colder than -38°C, which is typically above 7-8 km. Chen et al. (2000) instead used a pressure threshold of 440 hPa to separate clouds, which corresponds to an altitude of 6.3 km (standard atmosphere). The clouds in the range 6.3-7.5 km are missing in our study. Especially in this altitude range thick cirrus stratus (still with $\tau < 3.6$). Second,

10 although our criteria allow for thick clouds up to $\tau = 3.6$ at altitudes clearly above lower edge, they cut clouds once their lower edge gets warmer than -38°C, which is more likely for thicker clouds (see rounded edge in the red boxes in Figure 6). Third, while we count vertically distinct cirrus layers as separate clouds, the geostationary ISCCP weather satellites add the signal of vertically staggered layers, which increases $\tau$. Furthermore, it should be noted that satellite data reveal discrepancies amongst themselves (ISCCP, MISR, MODIS) with differences of 20-30% in coverage of cirrus with $\tau < 3.6$ (Marchand et al., 2010).

15 Finally, the distribution of thicker cirrus with $\tau > 0.3$ is zonally inhomogeneous, with clouds preferentially occurring at the continental east coasts.





In our study, we want to address only cirrus clouds and not mixed phase clouds. Therefore, we have chosen a conservative limit towards lower, thicker clouds. Also, a temperature-based selection criterion is a better for separating different cloud types than a pressure-based criterion, because temperature is the main microphysical parameter for cloud formation. As ISCCP is based on the analysis of weather satellite images, clouds still must have optical depths $\tau \geq 0.2$ in order to be reliably detected by such satellites (Rossow and Schiffer, 1999). Large uncertainties can also be traced to different approaches to partly cloudy pixels, which are 30 km x 30 km for ISCCP and are treated as homogeneous, i.e. either cloud free or filled with a thinned homogeneous cloud (Pincus et al., 2012)

The overcast and all sky $CRF_{NET}$ are significantly higher in Jülich than at Jungfraujoch, which is also clearly reflected in overcast and all sky optical depths found in the ISCCP data (Soden and Donner, 1994). This may be related to the frequent low-pressure systems and fronts rolling in from the northwest across the North Sea. The related cirrus clouds are weaken with distance from the coast.

The effect of the optically thicker clouds above Jülich compared to the Jungfraujoch is also evident in Fig. 8. The magenta lines indicate positive (i.e. warming) cirrus cloud radiative forcing (in $Wm^{-2}$) as a function of altitude and optical depth calculated by the model of Corti and Peter (2009) with mean temperature profiles from COSMO-2 (Jungfraujoch and Zürich) and COSMO-7 (Jülich) during the time period of our measurements. The blue isolines indicate negative (i.e. cooling) cirrus radiative forcing. A zero net effect, $CRF_{NET}=0$ is indicated by a cyan line. Color-coded is the occurrence frequency of the cirrus clouds measured at the different sites. The occurrence frequency is categorized by 40 logarithmically spaced bins in optical depth between $10^{-4}$ and 10 and 500-m bins in cloud top altitude. From Fig. 8 we clearly see that the cirrus clouds observed in this study have all a positive (warming) net radiative effect $CRF_{NET}$. It is important to note, that with the FLICA algorithm we do not find cirrostratus or cumulonimbus outflow clouds, i.e. no clouds with $\tau > 3.6$. Of course, such clouds do exist also above our measurement sites. However, such clouds always have lower edges warmer than -38°C and thus are not considered.

The pattern of cirrus cloud occurrence is quite similar above Jungfraujoch and Zürich, although the Jungfraujoch cirrus clouds show a slightly broader distribution in optical depths. Most cirrus layers are present at 11 km a.s.l. with optical depths between 0.01(0.04) and 0.2(0.4) above Jungfraujoch (in Zürich). Cirrus clouds above Jülich are frequent at altitudes between 8 and 12 km a.s.l. with optical depths ranging from 0.02 to 0.7. This wider distribution of high occurrence frequencies in altitude is likely related to the high frequency of frontal systems crossing Jülich. The lower $CRF_{NET}$ above Jungfraujoch is visible in the shift towards thinner clouds at Jungfraujoch as compared to the other two measurement sites. Due to the lower SNR over Zürich and Jülich at cirrus altitude, these two sites underestimate the amount of subvisible cirrus clouds as compared to Jungfraujoch.

Beside Chen et al. (2000) other studies indicate also a general net warming effect of cirrus clouds in the mid-latitudes. Oreopoulos and Rossow (2011) investigated the overall cloud radiative forcing based on a 24 year long data set from the International Satellite Cloud Climatology Project (ISCCP). For cases with frequent occurrence of high clouds, a positive net cloud





**Figure 8.** Cloud Radiative Forcing (CRF) in Wm$^{-2}$ for the different sites. Magenta/cyan/blue isolines: positive/zero/negative values in Wm$^{-2}$ from the CRF model (Corti and Peter, 2009). Color coding: Occurrence frequency of cirrus clouds as function of optical depth and cloud top altitude. First row: CRF$_{LW}$. Second row: CRF$_{SW}$. Third row: CRF$_{NET}$.





radiative forcing (warming) was obtained whereas it was negative (cooling) for cases with frequent occurrence of low-level clouds. This confirms our results where cirrus clouds create a positive net radiative forcing. A case study of Katagiri et al. (2010) found a cirrus cloud radiative forcing of 13.2 $Wm^{-2}$ at the Fukue observatory (32°N, Japan) by a combination of MODIS satellite and ground-based observations. This value is similar to what we found for Zürich and Jülich and obviously

larger than the respective value reported by Chen et al. (2000). Another study of Min et al. (2010) found a radiative forcing of cirrus of 36.5 $Wm^{-2}$ over China using also CALIOP and MODIS satellite data. The authors ascribe the high value to cirrus observations above the Tiberian plateau where very thick cirrus clouds with a mean optical depth of 1 are observed frequently. For the other parts of China lower values of 20 $Wm^{-2}$ are found. The radiative forcing of the lateral boundary of cirrus clouds observed with CALIOP is investigated by Li et al. (2014). The lateral boundary with optical depths less than 0.3 is found to

have still a radiative forcing of 10 $Wm^{-2}$. This value is similar to our mean overcast radiative forcing of Zürich and Jülich and demonstrates also the sensitivity of cirrus cloud inhomogeneity on cloud forcing as also found by Gu and Liou (2006). However, all these studies investigate single cases or different regions compared to our study. There is no study which investigates the cirrus radiative forcing over Europe.

In a study of Dupont et al. (2010) cirrus cloud observations over 2 years from the CALIPSO satellite lidar CALIOP are compared to four ground based lidar stations (two sites in the US and two in France) for their consistency of macrophysical and optical properties. They found larger discrepancies in the frequency distributions of cloud base, top and thickness. They point out that the significant part of the deviations can be attributed to different sampling (seasonal, irregular sampling of ground based stations, opaque low level clouds). However, they found that for high cirrus clouds the optical depth distribu-

tion $\tau > 0.1$ from ground stations and CALIOP is consistent within 10% using the same retrieval method. This shows that our optical depth distribution of all three stations is most likely not or only less affected by sampling issues of ground base lidar compared to satellite measurements. Further, we more closely examined the $CRF_{NET}$ categorized by their optical thickness $\tau$ as in Sassen and Cho (1992).

### 4.3 Influence of subvisible cirrus on the net radiative forcing by cirrus clouds

Subvisible cirrus clouds generally are not considered in numerical weather prediction models as their optical depths are considered to be too small. The overcast net radiative effect $CRF_{NET}$, divided into the categories defined by Sassen and Cho (1992), is shown in Fig. 9. We see that the subvisible cirrus clouds indeed have an effect on the net cirrus cloud radiative forcing $CRF_{NET}$. On average they contribute about 4 % of the total $CRF_{NET}$ of cirrus clouds at Jungfraujoch and in Zürich, and 3 % in Jülich. The maximal effect of 6 % is reached in Zürich during spring. As seen in Fig. 9, the thin and opaque cirrus clouds are the main

contributors to $CRF_{NET}$ of cirrus clouds above each of the three stations, both by roughly equal shares (with a small dominance of opaque clouds).

Jungfraujoch displays the lowest $CRF_{NET}$-values throughout the whole year. This pattern is also seen in the optical depths shown in Fig. 4a. Generally, thinner clouds are detected above Jungfraujoch than at the other two sites. This influences the





**Figure 9.** Cirrus radiative forcing under cloudy conditions, CRF$_{NET}$(overcast), for the different seasons on Jungfraujoch (blue), in Zürich (pink) and Jülich (green). Light shading: subvisible cirrus ($\tau < 0.03$). Medium shading: thin cirrus ($0.03 < \tau < 0.3$). Dark shading: opaque cirrus ($0.3 < \tau$).

CRF$_{NET}$: as more subvisible cirrus are observed at Jungfraujoch (cf. Table 2) and their contribution to CRF$_{NET}$ is smaller than the contribution by thin and opaque cirrus, this leads to a smaller CRF$_{NET}$ on Jungfraujoch. Summing up the percentages listed in Table 2, Jungfraujoch shows a fraction of 57 % thin and opaque cirrus while Zürich and Jülich both occurrence frequencies of 65 and 68% thin and opaque clouds, respectively. These different sums result in the CRFs listed in Table 3 (taking note of

5   the fact that the thresholds for the cirrus cloud categories (Sassen and Cho, 1992) are on a logarithmic scale).



## 5   Conclusions

. We have presented a cirrus climatology based on 13'000 hours of lidar measurements at the three different mid-latitude sites Jungfraujoch, Zürich and Jülich from 2010-2014. This extensive data set was evaluated using the newly developed FLICA algorithm, which combines a pixel-based cloud detection scheme with a classic lidar retrieval. With FLICA, the lidar data have been automatically evaluated. The retrieved backscatter coefficients are converted into extinction coefficients, which are corrected for multiple scattering to establish single scattering extinction and then converted into optical depths.

We find mean optical depths of 0.12 for the cirrus measured over Jungfraujoch and of 0.14 and 0.17, respectively, for Zürich and Jülich. While the cirrus coverage over Jungfraujoch and Jülich are almost equal, the amount of subvisible clouds detected over Jungfraujoch is significantly larger (cf. Table 2). Due to its unique location at 3580 m a.s.l., Jungfraujoch is an excellent site to measure subvisible cirrus clouds with a much improved SNR at cirrus altitude in comparison with the lower-lying stations. The mean cloud tops detected were located at 10.7 km in Zürich and on Jungfraujoch and at 10.3 km in Jülich, consistent with previous studies of Sassen and Campbell (2001) and Hoareau et al. (2013). Further, we have measured a temporal cirrus cloud coverage of 9-15 % with a mean value of 13 %. This is consistent with the evaluation of the global CALIPSO-measurements of Sassen et al. (2008).

The evaluated cirrus cloud properties are used together with the radiation model of Corti and Peter (2009) to estimate the cloud radiative forcing of the cirrus clouds. The optical depth as well as the cloud top temperature are the most important quantities determining the $CRF_{NET}$, and this dependence has been displayed in Fig. 8. Our results clearly confirm the warming effect of mid-latitude cirrus clouds with optical depths below 3, corroborating previous studies. Using the radiation model of Corti and Peter (2009), we find a net effect of 0.9 Wm$^{-2}$ for Jungfraujoch and 1.0/1.6 Wm$^{-2}$ for Zürich/Jülich. These values are larger by factors of 2-3 than the 50°N zonally averaged $CRF_{NET}$ derived by Chen et al. (2000) from satellite measurements in combination with a radiative transfer model. Even stronger deviations–but with opposite sign–are found for $CRF_{SW}$ and $CRF_{LW}$, where the zonally averaged data are higher than our CRF by up to one order of magnitude. This is due to different cloud definitions used by Chen et al. (2000) and us, and to the fact that the satellite-based zonal average includes regions with more pronounced thick cirrus (e.g. the continental east coasts).

The actual purpose of this work is the investigation of the thin ($0.03 < \tau < 0.3$) and subvisible $\tau < 0.03$) cirrus clouds, which remain undetected by satellites (requiring typically $\tau > 0.2$) and have so far not yet been systematically characterized in a climatological manner. The present study presents more than 13'000 hours of elastic backscatter lidar data, comprising more than 23'000 individual cirrus clouds. Of these clouds about 40% were subvisible, 50% thin, and 10% opaque cirrus. In terms of fraction of cloud coverage, subvisible cirrus were observed during about 6%, thin cirrus during about 7% and opaque cirrus during about 1.5% of the observation time. Seasonal variability in cirrus coverage shows characteristic autumn and spring maxima in agreement with satellite climatologies. Finally, in terms of cloud radiative forcing, all clouds discussed here show a





positive, i.e. warming, effect. We calculate that subvisibe cirrus contribute about 5%, thin cirrus about 45% and opaque cirrus about 50% of the total cirrus radiative forcing. In order to exert a negative forcing, i.e. a cooling effect, clouds need to be either optically much thicker or in altitude much lower, or both, but we excluded these clouds by demanding that the lower edge of the cloud needs to be colder than -38°C (cf. Fig. 8).

One important difference between the high ice clouds measured at Jungfraujoch compared to Jülich (with Zürich intermediate) is the possibility to measure thinner clouds above Jungfraujoch, which emphasizes the enhanced suitability of the high alpine measurement station to achieve a high signal-to-noise ratio. Reasons for this are that the objects of interest are closer (and the backscattered signals scales with the square of the distance) and that the polluted boundary layer stays often below the Jungfraujoch station. The Jungfraujoch data show that the lower detection limit in optical depth of a few times $10^{-5}$ is approached in a few cases, but by far the most subvisible cirrus stay clearly above $\tau = 10^{-4}$. We argue that this indicates that physical mechanisms prevent clouds to become and stay so thin for appreciable times. After formation, clouds will typically grow quickly and assume higher optical thicknesses. Conversely, evanescent clouds–once having become so thin–will evaporate quickly, not leaving much time for their detection. This leads us to speculate that the Jungfraujoch measurements enable us to explore the very onset of cirrus formation and to possibly learn from the lidar measurements about the relative importance of homogeneous and heterogeneous ice nucleation.

*Acknowledgements.* We are very grateful to Frank Wienhold for providing scripts for lidar evaluation and for input to an early version of the manuscript. We thank Uwe Weers, Marco Vecellio and Edwin Hausammann for technical support with the lidar. We are particularly grateful to Joan and Martin Fischer as well as Maria and Urs Otz for excellent local support at the Jungfraujoch. Thanks also to Albert Ansmann for very helpful scientific input, and to Andrew Huisman, Laura Revell as well as Silke Gegenbauer for proof reading of an early stage of the manuscript. This work has been funded by GAW-CH, the Swiss branch of the Global Atmosphere Watch (GAW) programme, coordinated by the GAW-CH Office at MeteoSwiss in Switzerland.



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
