# Peer review of "Climatological and radiative properties of mid-latitude cirrus clouds derived by automatic evaluation of lidar measurements"

_Atmospheric Chemistry and Physics, 2016_

## Short Comment (SC1) · 6 Feb 2016

This is a nice study of thin cirrus over 3 stations in the Alps and Northern Germany.

I have a few questions/comments/suggestions:

1) Which fraction of the thin cirrus originates from contrail cirrus? Liou et al. [1990], e.g., noted a strong increase of thin cirrus over Salt Lake City since about the late 1960's in correlation with increases in jet traffic. The stations are located in regions where line-shaped contrails are ubiquitous [Mannstein et al., 1999; Meyer et al., 2002]. The stations are located near the routes from London to the Near East or the routes from or across Paris to the Far East etc. (see contrail cover results and major traffic

routes in Fig. 7 in [Schumann, 2005]). Often aged contrail cirrus might have gotten advected from, e.g., the routes over Lyon to the central Alps. The observed optical depth is fully consistent with optical depth for contrail cirrus from other sources [Immler et al., 2008; Iwabuchi et al., 2012; Vázquez-Navarro et al., 2015]. The computed cover and RF values are consistent with contrail cirrus calculations [Schumann et al., 2015]. Hence, it is very likely that contrails contributed a large fraction to the observed thin cirrus. So far, your nice paper, not even mentions this possibility. I think, at least that needs to be changed.

2) How important for longwave radiative forcing (RF) from thin cirrus for otherwise clear sky is the water vapor in the atmosphere below the cirrus? The longwave RF of thin cirrus correlates far better with the brightness temperature of the atmosphere than with surface temperature, see Fig. 15.4 in [Schumann et al., 2012a].The brightness temperature is related to the outgoing longwave radiation (OLR) at top of the atmosphere, as available, e.g., from Numerical Weather Prediction(NWP) data, e.g. from COSMOS. Also: how important is the difference between Earth surface albedo and effective albedo of the Earth-Atmosphere system, e.g. when clouds are nearby the location of observations or when the mountains are snow covered or when there is any dust or haze (derivable from known solar direct radiation and from reflected shortwave radiation, RSR, also available from NWP data), as discussed in these papers? Perhaps you can quantify these effects?

3) Why not to test the differences between the nice and simple Corti&Peter parametrization and that which we developed in parallel (see my comment of May 2009 on the ACPD paper by Corti and Peter and [Schumann et al., 2012b])? The input needed (OLR and RSR) is available form COSMO and other NWP models. The model could be used to test the influence of various assumptions on particle habits and particle sizes [Markowicz and Witek, 2011]. The quantitative results may well change by 50 %, and hence change your conclusions.

4) Does the Lidar signal (e,g., depolarization) allow to discriminate, perhaps together

with other data, contrails from cirrus? Perhaps there are some ideas which could fit into your outlook?

References.

Immler, F., R. Treffeisen, D. Engelbart, K. Krüger, and O. Schrems (2008), Cirrus, contrails, and ice supersaturated regions in high pressure systems at northern mid latitudes, Atmos. Chem. Phys., 8, 1689–1699, doi:10.5194/acp-8-1689-2008.

Iwabuchi, H., P. Yang, K. N. Liou, and P. Minnis (2012), Physical and optical properties of persistent contrails: Climatology and interpretation, J. Geophys. Res., 117, D06215, doi:10.1029/2011JD017020.

Liou, K. N., S. C. Ou, and G. Koenig (1990), An investigation of the climatic effect of contrail cirrus. In: Air Traffic and the Environment – Background, Tendencies and Potential Global Atmospheric Effects. U. Schumann (Ed.), Lecture Notes in Engineering, Springer Berlin, 154-169.

Mannstein, H., R. Meyer, and P. Wendling (1999), Operational detection of contrails from NOAA-AVHRR data, Int. J. Remote Sensing, 20, 1641-1660, doi: 10.1080/014311699212650.

Markowicz, K. M., and M. Witek (2011), Sensitivity study of global contrail radiative forcing due to particle shape, J. Geophys. Res., 116, D23203, doi:10.1029/2011JD016345.

Meyer, R., H. Mannstein, R. Meerkötter, U. Schumann, and P. Wendling (2002), Regional radiative forcing by line-shaped contrails derived from satellite data, J. Geophys. Res., 107, ACL 17-11 - ACL 17-15, 10.1029/2001jd000426.

Schumann, U. (2005), Formation, properties and climate effects of contrails, Compt. Rend. Phys., 6, 549 - 565.

Schumann, U., K. Graf, H. Mannstein, and B. Mayer (2012a), Contrails: Visible aviation

induced climate impact, in Atmospheric Physics – Background - Methods - Trends, edited by U. Schumann, pp. 239-257, Springer, Berlin, Heidelberg, DOI: 10.1007/978-3-642-30183-4_15.

Schumann, U., B. Mayer, K. Graf, and H. Mannstein (2012b), A parametric radiative forcing model for contrail cirrus, J. Appl. Meteorol. Clim., 51, 1391-1406, doi: 10.1175/JAMC-D-11-0242.1.

Schumann, U., J. E. Penner, Y. Chen, C. Zhou, and K. Graf (2015), Dehydration effects from contrails in a coupled contrail-climate model, Atmos. Chem. Phys., 15, 11179-11199, doi:10.5194/acp-15-11179-2015.

Vázquez-Navarro, M., H. Mannstein, and S. Kox (2015), Contrail life cycle and properties from 1 year of MSG/SEVIRI rapid-scan images, Atmos. Chem. Phys., 15, 8739-8749, doi:10.5194/acp-15-8739-2015.
* * *

---

## Referee Comment (RC1) · Anonymous Referee #2 · 8 Feb 2016

Comments on 'Radiative properties of mid-latitude cirrus clouds derived by automatic evaluation of lidar measurements'
By Kienast-Sjögren et al.
Submitted to atmospheric chemistry and physics

In this article, an algorithm is developed (FLICA) to retrieve cirrus cloud properties based on lidar measurements at three stations in Europe. Using these retrievals, a cirrus climatology and cirrus radiative forcing in each station are presented. Differences of cirrus at three locations are discussed. Subviusal, thin and opaque cirrus are analyzed. Results are also compared to previous studies, and the differences with results by Chen et al. (2000) are particularly emphasized. This paper is generally completed and well written. My main comments/questions relate to the section of comparisons with previous studies, and methods to calculate ice cloud radiative forcing.

Specific comments.

1. Title of this study, 'Radiative properties of …', since a climatology of cirrus is also an important part of this study, is it better to say ' Climatological and radiative properties of…'

2. Aerosols
   How the algorithm distinguish aerosols and cirrus in this article? Cirrus clouds with small optical depth (e.g. tau<0.001) look more like aerosols?

3. Page 15. Line 5. How do you make sure such a small optical depth ($< 5 \times 10^{-4}$) is not resulted from noises or from aerosols?

4. Page 17, line 25, The mean solar zenith angle for three locations is 60°. However, JUL and JFJ are about 4 degrees latitude off, and thus the mean length of daytime (mean SZA) in the two locations should also be quite different, which will cause radiative flux biases. Have you ever check the differences?

5. Page 17, line 25, An albedo of 0.3 is globally average planetary albedo. The mean surface albedo is about 0.15 (Kiehl and Trenberth, 1997). Also, surface albedo varies from different locations and in different seasons. In particular, JFJ is located at a high altitude and has a cold climate. How many days of this location will be covered by snow in a year? Surface albedo covered by snow is large (> 50%).

6. Page 17. Line 29: The extinction coefficient can be derived from radar backscattering and then optical depth is obtained as shown in equation 7 in this paper. The tau values are used to calculate ice cloud radiative effect. Have you look at how different cirrus radiative effect will be if the extinction coefficient profile is used? The profiles characterize the vertical details of a cloud, which are more accurate to produce radiative fluxes (Chen 2000).

7. How the asymmetric factor and single-scattering albedo of the clouds are determined?

8. Section 4.2, Comparisons with previous results
   Several paragraphs in Section 4.2 describe how results in this study differ from Chen et al. 2000. I doubt the way of comparisons for the following reasons. 1) different definitions of cirrus as stated in this article, and thus radiative forcing of cirrus is different. 2) different resolutions, it is 280 km resolution in Chen et al. 2000, while in this study, lidar has a small field of view (1.5 mrad by 0.3 mrad); 3) time period is very different (four days in Chen's study and five years in this study); 4) More importantly, although the three stations are located around 50°N, comparing ice cloud properties to zonally average values at 50°N provided by Chen et al.2000 is unreasonable since ice clouds vary significantly around 50° (e.g. Sassen et al. 2008). I'm confused why this section is necessary. Do you want to check that your results are correct? If so, why not compare to CERES fluxes? Could you justify the reasons why such comparison is necessary in this study?

   Besides Chen et al. 2000, this article also lists a series of related studies in Section 4.2 (page 21, lines9-30). It may be better to move these paragraphs to introduction.

9. Page 25, line 11: ' cirrus clouds, which remain undetected by satellites (requiring typically tau>0.2)…', CALIPSO lidar can detect ice clouds with tau< 0.2. It would be better to revise the satellites as ' passive remote sensing'. It'll be better if a related reference added after tau> 0.2.

**References**

Chen, T., W. B. Rossow, and Y. Zhang, 2000: Radiative effects of cloud-type
Variations. J. Climate, 13, 264–286.

Kiehl, J. T., and K. E. Trenberth (1997), Earth's annual global mean energy budget, Bull. Am. Meteorol. Soc., 78, 197–208.

Sassen, K., Z. Wang, and D. Liu (2008), Global distribution of cirrus clouds from CloudSat/Cloud-Aerosol Lidar and Infrared Pathfinder Satellite Observations (CALIPSO) measurements, J. Geophys. Res., 113, D00A12, doi:10.1029/2008JD009972.

---

## Referee Comment (RC2) · L. Poole (Referee) · 25 Feb 2016

Referee comments on "Radiative properties of mid-latitude cirrus clouds derived by automatic evaluation of lidar measurements" by Erika Kienast-Sjögren et al.

General Comments

This is an interesting paper describing cirrus cloud occurrence frequencies, vertical distributions, and optical depths derived from lidar measurements at Zurich and Jungfraujoch Research Station, Switzerland, and Jülich, Germany. These results are compared with those from some earlier studies and are also used in a simple radiative transfer model to compute shortwave, longwave, and net cirrus radiative forcings. The paper is generally well written and the results are presented rather clearly. I do have a number of specific comments that the authors need to address before the paper is published in ACP.

Specific comments

The authors either are not aware of or have ignored some earlier papers describing ground-based lidar measurements of cirrus clouds obtained during the ECLIPS (Experimental Cloud Lidar Pilot Study) program. These papers include Platt et al., Bull.Amer.Met.Soc., 75, p.1635, 1994; Vaughan and Winker, Atmos.Res., 34, p.117, 1994; and Pal et al., J. Appl.Met., 34, p.2388, 1995. The authors should also mention how their new results compare with findings from these papers.

Pages 4-5: There is no discussion of the possibility of cross-talk between the co-polarized and cross-polarized channels of the lidar and the effect that might have on any results.

Page 5, line 26: The particulate lidar ratio can also be determined directly from high-spectral resolution lidar (HSRL) measurements.

Page 6, lines 17-18: It is not clear how the total uncertainty is computed. I don't think it should be the "sum" of the individual contributions as stated here. Is it the square root of the sum of the squares (RSS) of the individual contributions?

Page 7, line 1: Is the boxcar filter a moving average boxcar?

Page 7, line 24: What is meant by " a set of lidar ratios (5:5:40)"?

Page 7, line 31: Why is the temperature -38° C used to ensure pure ice clouds? Can the authors provide references?\

Page 12, Table 2, footnote (6): The text is confusing as written. Did the authors intend to say that relative uncertainties in their mean optical depths are comparable to "monthly mean values of 10-20% from ISCCP"?

Page 14, lines 9-19: It would be good if the authors did some statistical analysis on the optical depth distributions in Figure 4 and could state whether the various distributions are significantly different from a statistical point of view.

Page 20, line 3: From Table 3, I conclude that $CRF_{SW}$ at 50°N from ISCCP is about an order of magnitude than the present results, but the $CRF_{LW}$ at 50° N from ISCCP is only a factor of 1.5-3 larger.

Page 23, lines 8-10: I don't understand what is meant by "radiative forcing of the lateral boundary" of cirrus clouds? It would be good if the authors could provide a brief explanation.

Page 23, lines 22-23: I don't understand the last sentence of this paragraph. What did the "close examination of $CRF_{NET}$" with respect to cloud $\tau$ show?

Technical Corrections

Page 1, line 2: It would be better to say that cirrus "…affect the water vapor budget …" not determine it.

Page 1, line 15: Reword to say "… thus enabling lidar measurements of higher ..."

Page 2, line 3: The word "subvisible" is misspelled.

Page 20, line 9: Reword sentence to say "Cirrostratus clouds with $\tau<3.6$ occur particularly in this altitude range."

Page 26, line 1: The word "subvisible" is misspelled again.

---

## Author Response (AR1)

**Answer to Dr. Lamont Poole**

The authors are grateful for the time and thought that Lamont Poole put into the review and comments regarding our paper. We incorporate most of the comments into the revised manuscript, which has led to substantial improvements. Detailed responses to all comments follow below. The original comments from Lamont Poole are in italics and our responses as well as changes in the manuscript in plain text.

This is an interesting paper describing cirrus cloud occurrence frequencies, vertical distributions, and optical depths derived from lidar measurements at Zurich and Jungfraujoch Research Station, Switzerland, and Jülich, Germany. These results are compared with those from some earlier studies and are also used in a simple radiative transfer model to compute shortwave, longwave, and net cirrus radiative forcings. The paper is generally well written and the results are presented rather clearly. I do have a number of specific comments that the authors need to address before the paper is published in ACP.

**Specific comments**

The authors either are not aware of or have ignored some earlier papers describing groundbased lidar measurements of cirrus clouds obtained during the ECLIPS (Experimental Cloud Lidar Pilot Study) program. These papers include Platt et al., Bull.Amer.Met.Soc., 75, p.1635, 1994; Vaughan and Winker, Atmos.Res., 34, p.117, 1994; and Pal et al., J. Appl.Met., 34, p.2388, 1995. The authors should also mention how their new results compare with findings from these papers.

**Response:**

Thanks for your comment. We were not aware of these particular publications. We have added some of the results from these papers to our manuscript and compared them to our measurements.

**Changes in the revised manuscript are marked in blue:**

Lines 24 on page 2:

Lidar (LIght Detection And Ranging) measurements can be used to establish long time series of aerosol or cloud measurements (e.g., Platt et al., 1994).

**Lines 2-3 on page 12:**

agreeing well with the CALIPSO measurements discussed by Sassen et al. (2008) and being slightly smaller than the 18-19% measured during the ECLIPS campaign by Winker and Vaughan (1994).

**Lines 11-12 on page 15:**

These  $\tau$  values agree well with the ECLIPS-campaign (Pal et al., 1995), where most detected cirrus clouds had optical depths smaller than 0.1.

Pages 4-5: There is no discussion of the possibility of cross-talk between the co-polarized and cross-polarized channels of the lidar and the effect that might have on any results.

**Response:**

We have added information about the possibility of cross-talk to the manuscript.

**Changes in the revised manuscript on page 5, lines 1-6 are marked in blue:**

We assume an ideal lidar system, which means that there is no cross-talk between the copolarized and the cross-polarized channels. Rolf (2012) has examined this for the lidar used in Jülich. He found that for the parallel detector every 2000th detected photon is actual perpendicular polarized and for the perpendicular detector about every 570 detected photon is parallel polarized. While this justifies our assumption of an ideal system for the Jülich lidar, we found considerable cross-talk in the Swiss lidar, depending on certain maintenance conditions. However, cross-talk influences in particular the perpendicular channel which we use mainly for cloud detection but not for optical depth retrieval.

Page 5, line 26: The particulate lidar ratio can also be determined directly from high-spectral resolution lidar (HSRL) measurements.

**Response:**

Thanks for this remark, we have added this information to the manuscript.

**Changes in the revised manuscript on page 6, lines 2-3 are marked in blue:**

It can be obtained directly from Raman lidars that allow for an independent measurement of particle extinction and backscatter coefficients (Cooney, 1972; Giannakaki et al., 2007; Radlach et al., 2008; Reichardt et al., 2002; Achtert et al., 2013) as well as from high-spectral resolution lidar (HSRL) measurements (e.g., Burton et al., 2012).

Page 6, lines 17-18: It is not clear how the total uncertainty is computed. I don't think it should be the "sum" of the individual contributions as stated here. Is it the square root of the sum of the squares (RSS) of the individual contributions?

**Response:**

We combined the uncertainties in such a way that we assess the "worst case" of uncertainty. We have added this information to the paper with the remark, that a Gaussian error would be smaller.

**Changes in the revised manuscript on page 6, lines 25-27 are marked in blue:**

To assess the total, maximum uncertainty, we combine the individual contributions to provide an upper bound of the uncertainty. We calculate the largest possible error, which usually is larger than the error calculated by a Gaussian error (square root of the sum of the squares (RSS) of the individual contributions).

**Page 7, line 1: Is the boxcar filter a moving average boxcar?**

**Response:**

Yes, this information has been added to the manuscript.

**Changes in the revised manuscript on page 7, line 9 are marked in blue:**

"further smoothed using a moving average boxcar filter"

Page 7, line 24: What is meant by "a set of lidar ratios (5:5:40)"?

**Response:**

We use lidar ratios between 5 and 40, with steps of 5 in between. The sentence has been changed in the manuscript.

Changes in the revised manuscript on page 7, line 32 are marked in blue:

"lidar ratios between 5 and 40 sr, in steps of 5 sr ..."

Page 7, line 31: Why is the temperature -38° C used to ensure pure ice clouds? Can the authors provide references?

**Response:**

We chose this threshold as it is the threshold for homogeneous nucleation to take place. We ensure that we detect pure ice clouds and exclude the mixed-phase clouds. We have added references to the manuscript.

Changes in the revised manuscript on page 8, lines 8-9 are marked in blue:

"Temperature has to be lower than -38°C (e.g., Pruppacher and Klett, 1997; Koop et al., 2000; Krämer et al., 2016) to ensure pure ice clouds and avoid detecting mixed-phase clouds (this..."

Page 12, Table 2, footnote (6): The text is confusing as written. Did the authors intend to say that relative uncertainties in their mean optical depths are comparable to "monthly mean values of 10-20% from ISCCP"?

**Response:**

The mean values of tau compare reasonably well with monthly mean tau values of 0.1-0.2 from ISCCP. We have changed the sentence accordingly in the manuscript.

Changes in the revised manuscript on page 13, table 2, footnote 6 are marked in blue:

"Mean values of  $\tau$  compare reasonably well with monthly mean  $\tau$  values of 0.1-0.2 from ISCCP..."

Page 14, lines 9-19: It would be good if the authors did some statistical analysis on the optical depth distributions in Figure 4 and could state whether the various distributions are significantly different from a statistical point of view.

**Response:**

We have used a Wilcoxon rank sum test to test this. We find p-values smaller than 1e-9, which clearly indicates that the distributions are significantly different.

**Changes in the revised manuscript on page 15, lines 12-13 are marked in blue:**

A Wilcoxon Rank sum test reveals that the optical depth distributions of the different sites are significantly different from each other.

Page 20, line 3: From Table 3, I conclude that CRFSW at 50°N from ISCCP is about an order of magnitude than the present results, but the CRFLW at 50° N from ISCCP is only a factor of 1.5-3 larger.

**Response:**

Thank you for pointing this out. We have provided this information to the manuscript and also added some explanations.

**Changes in the revised manuscript on page 21, lines 15-16 and page 22, line 1 are marked in blue:**

"1.5-3 times larger radiative forcing in the longwave,  $CRF_{LW}$ , and one order of magnitude larger radiative forcing in the shortwave,  $CRF_{SW}$ . The difference in the  $CRF_{SW}$  can only be explained in terms of a..."

Page 23, lines 8-10: I don't understand what is meant by "radiative forcing of the lateral boundary" of cirrus clouds? It would be good if the authors could provide a brief explanation.

**Response:**

In this cited paper the authors (Li et al., 2014) discussed the radiative effect of observed cirrus cloud edges. With lateral boundary they describe the first 10 km horizontal from outside cirrus clouds with optical depth of 0 towards small optical depth less than 0.3. In this transition region, they found still a  $CRF_{LW}$  of 10 W/m2.

This information has been added to the manuscript.

**Changes in the revised manuscript on page 23, lines 29-31 are marked in blue:**

The radiative effect of observed cirrus cloud edges is discussed. In the transition region of large cirrus, defined as their optically thin rim ( $\tau < 0.3$ ), which is often missed by satellite passive optical sensors such as MODIS, the CRFLW found to be still substantial (~ 10 Wm-2).

Page 23, lines 22-23: I don't understand the last sentence of this paragraph. What did the "close examination of CRFNET" with respect to cloud  $\tau$  show?

**Response:**

Thank you for pointing to this ambiguity. We have removed the sentence, since it belongs to the paragraph below and is repeated there.

Changes in the revised manuscript on page 25, lines 8-9 are marked in blue:

The sentence was removed.

**Technical Corrections**

Page 1, line 2: It would be better to say that cirrus "...affect the water vapor budget ..." not determine it.

Page 1, line 15: Reword to say "... thus enabling lidar measurements of higher ..."

Page 2, line 3: The word "subvisible" is misspelled.

Page 20, line 9: Reword sentence to say "Cirrostratus clouds with  $\tau < 3.6$  occur particularly in this altitude range."

Page 26, line 1: The word "subvisible" is misspelled again.

Response:

These corrections have all been implemented.

**Answer to Anonymous Referee 2**

The authors are grateful for the time and thought that Anonymous Referee 2 put into the review and comments regarding our paper. We incorporate most of those comments into our revised manuscript, which has led to substantial improvements. Detailed responses to all comments follow below. The original comments from Anonymous Referee 2 are in italics and our responses as well as changes in the manuscript in plain text.

In this article, an algorithm is developed (FLICA) to retrieve cirrus cloud properties based on lidar measurements at three stations in Europe. Using these retrievals, a cirrus climatology and cirrus radiative forcing in each station are presented. Differences of cirrus at three locations are discussed. Subvisual, thin and opaque cirrus are analyzed. Results are also compared to previous studies, and the differences with results by Chen et al. (2000) are particularly emphasized. This paper is generally completed and well written. My main comments/questions relate to the section of comparisons with previous studies, and methods to calculate ice cloud radiative forcing.

**Specific comments.**

1. Title of this study, 'Radiative properties of ...', since a climatology of cirrus is also an important part of this study, is it better to say 'Climatological and radiative properties of...'

**Response:**

This is an excellent idea. We have changed the title accordingly.

**2. Aerosols:**

How the algorithm distinguish aerosols and cirrus in this article? Cirrus clouds with small optical depth (e.g. tau<0.001) look more like aerosols?

**Response:**

FLICA uses a criterion for depolarization to help preventing that aerosols are classified as cirrus clouds. As we have already stated in the manuscript, to ensure that the highest cirrus clouds observed above Jungfraujoch were not volcanic ash particles (which would also depolarize the backscattered light), we have examined satellite measurements and found no indication for volcanic influence. Furthermore, our temperature threshold of -38°C excludes most aerosols because most aerosol layers are located below 6 km. Even though this combination of criteria might not be perfect, we believe that cirrus clouds are clearly distinguished from aerosols in the very most cases.

**Changes in the manuscript:**

No changes have been applied to the manuscript.

**3. Page 15. Line 5. How do you make sure such a small optical depth (< 5x10-4) is not resulted from noises or from aerosols?**

**Response:**

Areas of  $3 \times 3$  pixels are examined, with the pixel to be checked for cloudiness in the center. At least 8 of the 9 examined pixels have to have a volume depolarization larger than 0.007(0.006) and a BSR larger than 1.08(1.03) for day(night)-time measurements. As mentioned above, the output of the cloud detection scheme was in addition visually inspected for individual days and was found not to show any apparent artifacts (such as aerosol layers or noise).

**Changes in the manuscript:**

No changes have been applied to the manuscript.

4. Page 17, line 25, The mean solar zenith angle for three locations is 60°. However, JUL and JFJ are about 4 degrees latitude off, and thus the mean length of daytime (mean SZA) in the two locations should also be quite different, which will cause radiative flux biases. Have you ever check the differences?

**Response:**

Thank you for pointing out this issue. The differences in latitude between the three stations have two consequences. First, the length of daytime is different for the stations. This value is however taken directly from the measurements (see line 28 on page 19). Second, the differences in the solar angle affect the incident radiation. The overall difference in the incident solar radiation is 6%. Nevertheless, the radiation model of Corti & Peter has an accuracy that "is typically better than 20% when comparing with the accurate results from the Fu and Liou (1992, 1993) radiative transfer model", thus we consider this error to be of minor relevance for the model results. In addition, addressing as well item 5 below, we want to examine the radiative properties of the cirrus clouds as a function of optical depth and temperature under otherwise comparable conditions.

**Changes in the manuscript:**

No changes have been applied to the manuscript.

5. Page 17, line 25, An albedo of 0.3 is globally average planetary albedo. The mean surface albedo is about 0.15 (Kiehl and Trenberth, 1997). Also, surface albedo varies from different locations and in different seasons. In particular, JFJ is located at a high altitude and has a cold climate. How many days of this location will be covered by snow in a year? Surface albedo covered by snow is large (> 50%).

**Response:**

Jungfraujoch lies on top of a glacier and is all year covered by snow. We have made calculations using albedos of snow (0.65) for Jungfraujoch. If we use a snow-albedo, the radiative effect of the cirrus clouds disappear as all radiation is scattered back by the snow surface. As already mentioned in reply to item Ulrich Schumann, we chose a value of 0.3 to demonstrate the global average effect of cirrus clouds such as those detected above the three locations.

**Changes in the manuscript:**

No changes have been applied to the manuscript.

6. Page 17. Line 29: The extinction coefficient can be derived from radar backscattering and then optical depth is obtained as shown in equation 7 in this paper. The tau values are used to calculate ice cloud radiative effect. Have you look at how different cirrus radiative effect

will be if the extinction coefficient profile is used? The profiles characterize the vertical details of a cloud, which are more accurate to produce radiative fluxes (Chen 2000).

**Response:**

For this work, we have chosen to focus on the model of Corti and Peter, which uses the optical depth and cloud temperature as input (besides the surface albedo. This model has an accuracy better than 20% as compared to the model of Fu and Liou. We added this information to the manuscript.

Sentence added to the manuscript on page 19, line 13-14 in blue:

The accuracy of Corti and Peter (2009) is better than 20 % in comparison with the Fu-Liou model.

**7. How the asymmetric factor and single-scattering albedo of the clouds are determined?**

**Response:**

The asymmetry factor determines the forward scattering of a cloud and thus also the reflectivity. The reflectivity is approximated depending on optical depth and a fixed value as a result of radiative transfer calculations using the Fu-Liou code. For more detail see Corti and Peter, 2009. p. 5755.

Our value for single scattering albedo is  $\omega_0 = 0.55$ , which is a realistic estimate for longwave radiation (Stephens et al., 1990). The exact value is not decisive for our parameterization, since variations of 10 % in this parameter increases the mean error of CRFLW in comparision with radiative transfer calculations only by about 1%. Again, for more detail see Corti and Peter, 2009. p. 5754.

**Changes in the manuscript:**

No changes have been applied to the manuscript.

**8. Section 4.2, Comparisons with previous results**

Several paragraphs in Section 4.2 describe how results in this study differ from Chen et al. 2000. I doubt the way of comparisons for the following reasons. 1) different definitions of cirrus as stated in this article, and thus radiative forcing of cirrus is different. 2) different resolutions, it is 280 km resolution in Chen et al. 2000, while in this study, lidar has a small field of view (1.5 mrad by 0.3 mrad); 3) time period is very different (four days in Chen's study and five years in this study); 4) More importantly, although the three stations are located around 50°N, comparing ice cloud properties to zonally average values at 50°N provided by Chen et al.2000 is unreasonable since ice clouds vary significantly around 50° (e.g. Sassen et al. 2008). I'm confused why this section is necessary. Do you want to check that your results are correct? If so, why not compare to CERES fluxes? Could you justify the reasons why such comparison is necessary in this study?

**Response:**

We wanted to compare our data with similar measurements. There are only few publications assessing the radiative effect of mid-latitude cirrus clouds and the paper by Chen et al. is one of them. We are aware (and mention extensively in section 4.2) that it is very difficult to compare our and Chen's results due to the reasons you mention. Concerning a comparison

with CERES fluxes, we are not aware of a publication discussing cloud radiative forcings derived from the CERES data for mid-latitude cirrus. Conversely, establishing CRF for cirrus from CERES data is beyond the scope of the present investigation.

**Changes in the manuscript:**

No changes have been applied to the manuscript.

Besides Chen et al. 2000, this article also lists a series of related studies in Section 4.2 (page 21, lines9-30). It may be better to move these paragraphs to introduction.

**Response:**

After consultation with all co-authors we decided to keep the information at its origin position in the article as we think it is most suitable where it is.

9. Page 25, line 11: ' cirrus clouds, which remain undetected by satellites (requiring typically tau>0.2)...', CALIPSO lidar can detect ice clouds with tau< 0.2. It would be better to revise the satellites as ' passive remote sensing'. It'll be better if a related reference added after tau> 0.2.

**Response:**

We agree and have adapted this in the manuscript, reading now "passive remote-sensing satellites".

Changes in the manuscript on page 28, line 4 in blue:

"by passive remote-sensing satellites"

**Reply to the discussion comment submitted by Prof. Ulrich Schumann**

The authors are grateful for the time and thought that Ulrich Schumann put into the comments regarding our paper. Subsequently we show the original comments from Ulrich Schumann in italics and our responses as well as changes in the manuscript in plain text.

We appreciate that Mr. Schumann highlights the occurrence of contrail cirrus as an important point, especially over the European continent, which is subject of strong air traffic. Concerning this aspect, we of course have to admit that the developed data analysis scheme, FLICA, is not intended to separate contrail from natural cirrus. Such a separation would require – according to our understanding – the incorporation of side information from either detailed flight maps or satellite imagery to identify periods of strong contrail formation. Such an analysis was outside the scope of this study. In a possible future follow-up study a separation from natural and contrail cirrus could become an important component.

We decided to mention the topic in the conclusions of the manuscript, highlighting the potential of such long-term datasets for the characterization of anthropogenic effects on cirrus cloud properties. We also incorporated some of the references provided by Prof. Schumann.

**This is a nice study of thin cirrus over 3 stations in the Alps and Northern Germany.**

1) Which fraction of the thin cirrus originates from contrail cirrus? Liou et al. [1990], e.g., noted a strong increase of thin cirrus over Salt Lake City since about the late 1960's in correlation with increases in jet traffic. The stations are located in regions where line-shaped contrails are ubiquitous [Mannstein et al., 1999; Meyer et al., 2002]. The stations are located near the routes from London to the Near East or the routes from or across Paris to the Far East etc. (see contrail cover results and major traffic routes in Fig. 7 in [Schumann, 2005]). Often aged contrail cirrus might have gotten advected from, e.g., the routes over Lyon to the central Alps. The observed optical depth is fully consistent with optical depth for contrail cirrus from other sources [Immler et al., 2008; Iwabuchi et al., 2012; Vázquez-Navarro et al., 2015]. The computed cover and RF values are consistent with contrail cirrus calculations [Schumann et al., 2015]. Hence, it is very likely that contrails contributed a large fraction to the observed thin cirrus. So far, your nice paper, not even mentions this possibility. I think, at least that needs to be changed.

**Response:**

We find clear evidence of detected contrails only on one of the days. However, an uncertain number of the detected cirrus clouds may have evolved from contrails. How many is impossible to say without further, elaborate calculations. Especially, aged contrails in a water vapor supersaturated environment are hard to discriminate from natural cirrus clouds when only a stationary lidar is available. Due to turbulent mixing of contrail air with surrounding air masses and further growth of contrail ice crystals their microphysical properties become indistinguishable from those of natural cirrus. We have added a paragraph plus some references about this.

**Paragraph added in revised manuscript on page 27, lines 11-20 in blue:**

Owing to the central location of the three measurement sites in Europe, a significant fraction of the thin cirrus observed within the present study might actually have originated from contrails. Clear indications for the occurrence of contrails were found on at least one day, given the optically and geometrically very thin cirrus. Liou et al. (1990) noted an increase of thin cirrus with increases in jet traffic. Our measurement sites are located in a region, where line-shaped contrails are ubiquitous (Mannstein et al., 1999; Meyer et al., 2002) as many flight routes cross this area. The observed optical depths are consistent with optical depths of

contrail cirrus (Immler et al., 2008; Iwabuchi et al., 2012; Vázquez-Navarro et al., 2015). Furthermore, the cirrus cloud cover determined in the present study is consistent with contrail cirrus calculations by Schumann et al. (2015). Therefore, it is likely that contrails contributed a fraction of the observed cirrus. The determination of the actual contribution of contrails to the cirrus cloud dataset is, however, not subject of this study, considering that the applied data analysis algorithm FLICA cannot distinguish natural and contrail cirrus.

2) How important for longwave radiative forcing (RF) from thin cirrus for otherwise clear sky is the water vapor in the atmosphere below the cirrus? The longwave RF of thin cirrus correlates far better with the brightness temperature of the atmosphere than with surface temperature, see Fig. 15.4 in [Schumann et al., 2012a]. The brightness temperature is related to the outgoing longwave radiation (OLR) at top of the atmosphere, as available, e.g., from Numerical Weather Prediction(NWP) data, e.g. from COSMOS. Also: how important is the difference between Earth surface albedo and effective albedo of the Earth-Atmosphere system, e.g. when clouds are nearby the location of observations or when the mountains are snow covered or when there is any dust or haze (derivable from known solar direct radiation and from reflected shortwave radiation, RSR, also available from NWP data), as discussed in these papers? Perhaps you can quantify these effects?

**Response:**

We have added some remarks about the brightness temperature and its relation to the longwave radiative forcing to the manuscript and added the suggested reference.

As also mentioned in our response to reviewer 2, we chose a value of 0.3 for the underlying albedo to demonstrate, which radiative effect the detected cirrus would have, if they were located above the "mean" of the planet. Jungfraujoch is located on top of a glacier and is all year covered by snow. We have made calculations using albedos of snow (0.65) for Jungfraujoch. In that case, the radiative effect of the cirrus clouds disappears as all radiation is scattered back by the snow surface.

Changes in revised manuscript on page 19, lines 18-20 in blue:

The  $CRF_{LW}$  further correlates well with the brightness temperature of the atmosphere, which is related to the outgoing longwave radiation at top of atmosphere (Schumann et al., 2012a). This correlation has not been considered in the model of Corti and Peter (2009).

3) Why not to test the differences between the nice and simple Corti&Peter parametrization and that which we developed in parallel (see my comment of May2009 on the ACPD paper by Corti and Peter and [Schumann et al., 2012b])? The input needed (OLR and RSR) is available form COSMO and other NWP models. The model could be used to test the influence of various assumptions on particle habits and particle sizes [Markowicz and Witek, 2011]. The quantitative results may well change by50 %, and hence change your conclusions.

**Response:**

Thank you for this interesting remark. We have added a remark on this in the manuscript and provided the suggested references, but refrain from performing additional computations at this stage of the paper and leave this to potential follow-up projects.

Changes in revised manuscript on page 27, lines 33-35 and page 28, line 1 in blue:

Besides the radiation model of Corti and Peter (2009) used for this study, other approaches exist that can be used to investigate the effect of other cloud properties besides optical depth on the cirrus radiative forcing. For instance, the radiation model of Schumann et al. (2012b) could be used to test the influence of various assumptions on particle habits and particle sizes (Markowicz and Witek, 2011).

4) Does the Lidar signal (e,g., depolarization) allow to discriminate, perhaps together with other data, contrails from cirrus? Perhaps there are some ideas which could fit into your outlook?

**Response:**

We have not examined this so far. We think that it might be possible to distinguish fresh contrails from cirrus clouds as the contrails have a large number of small particles that are rather round due to the rapid cooling they were exposed to (Jensen, 1998). This would result in different depolarization values than for natural cirrus clouds. In a supersaturated environment, contrails can stay persistent for a long time and are more similar to natural cirrus clouds after growth. In a subsaturated environment the contrails will evaporate very quickly.

**Changes in revised manuscript:**

No changes have been made in the manuscript.

References implemented in the revised version of the paper:

Immler, F., R. Treffeisen, D. Engelbart, K. Krüger, and O. Schrems (2008), Cirrus, contrails, and ice supersaturated regions in high pressure systems at northern mid latitudes, Atmos. Chem. Phys., 8, 1689–1699, doi:10.5194/acp-8-1689-2008.

Iwabuchi, H., P. Yang, K. N. Liou, and P. Minnis (2012), Physical and optical properties of persistent contrails: Climatology and interpretation, J. Geophys. Res., 117, D06215, doi:10.1029/2011JD017020.

Liou, K. N., S. C. Ou, and G. Koenig (1990), An investigation of the climatic effect of contrail cirrus. In: Air Traffic and the Environment – Background, Tendencies and Potential Global Atmospheric Effects. U. Schumann (Ed.), Lecture Notes in Engineering, Springer Berlin, 154-169.

Mannstein, H., R. Meyer, and P. Wendling (1999), Operational detection of contrails from NOAA-AVHRR data, Int. J. Remote Sensing, 20, 1641-1660, doi: 10.1080/014311699212650.

Markowicz, K. M., and M. Witek (2011), Sensitivity study of global contrail radiative forcing due to particle shape, J. Geophys. Res., 116, D23203, doi:10.1029/2011JD016345.

Meyer, R., H. Mannstein, R. Meerkötter, U. Schumann, and P. Wendling (2002), Regional radiative forcing by line-shaped contrails derived from satellite data, J. Geophys. Res., 107, ACL 17-11 - ACL 17-15, 10.1029/2001jd000426.

Schumann, U., K. Graf, H. Mannstein, and B. Mayer (2012a), Contrails: Visible aviation induced climate impact, in Atmospheric Physics – Background - Methods - Trends, edited by U. Schumann, pp. 239-257, Springer, Berlin, Heidelberg, DOI: 10.1007/978-

3-642-30183-4\_15.

Schumann, U., B. Mayer, K. Graf, and H. Mannstein (2012b), A parametric radiative forcing model for contrail cirrus, J. Appl. Meteorol. Clim., 51, 1391-1406, doi: 10.1175/JAMC-D-11-0242.1.

Schumann, U., J. E. Penner, Y. Chen, C. Zhou, and K. Graf (2015), Dehydration effects from contrails in a coupled contrail-climate model, Atmos. Chem. Phys., 15, 11179-11199, doi:10.5194/acp-15-11179-2015.

Vázquez-Navarro, M., H. Mannstein, and S. Kox (2015), Contrail life cycle and properties from 1 year of MSG/SEVIRI rapid-scan images, Atmos. Chem. Phys., 15, 8739-8749, doi:10.5194/acp-15-8739-2015.

**Climatological and radiative properties of mid-latitude cirrus clouds derived by automatic evaluation of lidar measurements**

Erika Kienast-Sjögren1,2, Christian Rolf3, Patric Seifert4, Ulrich K. Krieger1, Bei P. Luo1,5, Martina Krämer3, and Thomas Peter1

1Institute for Atmospheric and Climate Science, ETH Zurich, Switzerland
2Now at: Fed. Office of Meteorology and Climatology, MeteoSwiss, Zurich Airport, Operation Center 1, CH-8058 Zurich, Switzerland
3Institute for Energy and Climate Research, Stratosphere, Forschungszentrum Jülich, Jülich, Germany
4Institute for Tropospheric Research (TROPOS), Leipzig, Germany
5Physical Meteorological Observatory Davos, PMOD WRC, CH-7260 Davos, Switzerland

Correspondence to: Erika Kienast-Sjögren (Erika.Kienast@meteoswiss.ch)

**Abstract.** Cirrus, i.e. high thin clouds that are fully glaciated, play an important role in the Earth's radiation budget as they interact with both long- and shortwave radiation and determine affect the water vapor budget of the upper troposphere and stratosphere. Here, we present a climatology of mid-latitude cirrus clouds measured with the same type of ground-based lidar at three mid-latitude research stations: at the Swiss high alpine Jungfraujoch station (3580 m a.s.l.), in Zürich (Switzerland, 510

- 5 m a.s.l.) and in Jülich (Germany, 100 m a.s.l.). The analysis is based on 13'000 hours of measurements from 2010 2014. To automatically evaluate this extensive data set, we have developed the "Fast LIdar Cirrus Algorithm" (FLICA), which combines a pixel-based cloud-detection scheme with the classic lidar evaluation techniques. We find mean cirrus optical depths of 0.12 on Jungfraujoch and of 0.14 and 0.17 in Zürich and Jülich, respectively.
- 10 Above Jungfraujoch, subvisible cirrus clouds ( $\tau < 0.03$ ) have been observed during 7% of the observation time, whereas above Zürich and Jülich significantly less. From Jungfraujoch, clouds with  $\tau < 10^{-3}$  can be observed three times more often than over Zürich and Jülich, and clouds with  $\tau < 2 \times 10^{-4}$  even ten times more often. Above Jungfraujoch, cirrus have been observed to altitudes of 14.4 km a.s.l., whereas only to about 1 km lower at the other stations. These features highlight the advantage of the high-altitude station Jungfraujoch, which is often in the free troposphere above the polluted boundary layer,
- 15 thus allowing to perform enabling lidar measurements of thinner and higher clouds. In addition, the measurements suggest a change in cloud morphology at Jungfraujoch above  $\sim$ 13 km, possibly because high particle number densities form in the observed cirrus clouds, when many ice crystals nucleate in the high supersaturations following rapid uplifts in lee waves above mountainous terrain.
- 20 The retrieved optical properties are used as input for a radiative transfer model to estimate the net cloud radiative forcing,  $CRF_{NET}$ , for the analysed cirrus clouds. All cirrus detected here have a positive  $CRF_{NET}$ . This confirms that these thin, high cirrus have a warming effect on the Earth's climate, whereas cooling clouds typically have lower cloud edges too low in altitude

to satisfy the FLICA criterion of temperatures below -38° C. We find  $CRF_{NET} = 0.9 \text{ Wm}^{-2}$  for Jungfraujoch and 1.0 Wm-2 (1.7 Wm-2) for Zürich (Jülich). Further, we calculate that subvisible cirrus ( $\tau < 0.03$ ) contribute about 5%, thin cirrus (0.03  $< \tau < 0.3$ ) about 45% and opaque cirrus ( $0.3 < \tau$ ) about 50% of the total cirrus radiative forcing.

**1 Introduction**

- 5 One of the large challenges in climate modeling, characterized by a low level of scientific understanding, are clouds and their effects on climate (Dessler and Yang, 2003; Solomon et al., 2007; Boucher et al., 2013). This concerns also the microphysical processes leading to cirrus formation. These processes are subject to uncertainties in the understanding and parametrization of homogeneous and heterogeneous nucleation (e.g., Cirisan et al., 2014). For any specific cloud scene, unless there are in situ measurements, there is either no or incomplete knowledge of the number of ice nuclei (IN), the intensity of small-scale temper-
- 10 ature fluctuations or the corresponding accurate values of upper tropospheric humidity (e.g., Ickes et al., 2015; Kienast-Sjögren et al., 2015).

Cloud properties such as cloud particle number, size and ice particle shape determine ice water content and optical depth, which together with the temperature of the cirrus cloud top determines whether the net cloud radiative forcing,  $CRF_{NET}$ , is

- 15 positive or negative, i.e. whether a particular cirrus cloud is warming or cooling (Platt and Harshvardhan, 1988; Ebert and Curry, 1992; Lin et al., 1998a; Chen et al., 2000; Corti and Peter, 2009). The fact that liquid clouds contain spherical particles helps estimating their microphysical and radiative properties. Conversely, the different shapes and orientations (Pruppacher and Klett, 1997) of ice particles affect the extinction of light, complicating the estimation of the cirrus climate effect (Fu and Liou, 1993; Liou, 2002). Previous studies of the radiative effect of cirrus (e.g., Chen et al., 2000; Fusina et al., 2007; Cziczo
- and Froyd, 2014) have identified a range of several watts per square meter  $(Wm^{-2})$  depending on the ice crystal number in a cirrus as compared to having an ice-free supersaturated region.

Lidar (LIght Detection And Ranging) measurements can be used to establish long time series of aerosol or cloud measurements (e.g., Platt et al., 1994). From the co- and cross-polarized components of the backscattered light the profile of the
depolarization ratio can be obtained providing information about the sphericity of the retrieved particles, and thus on their liquid or solid state. Several lidar stations have applied their measurements of elastically backscattered light to investigate the properties of mid-latitude cirrus clouds. See Table 1 for an overview.

Here we present a cirrus cloud climatology based on 13'000 hours of lidar measurements from three mid-latitude sites;
Jungfraujoch, Zürich and Jülich. The lidar technique is briefly described in Section 2.1. In Section 2.2, the newly developed evaluation algorithm FLICA is presented. Using FLICA we are able to analyze extensive lidar measurements automatically. The climatology of this data is presented in Section 3. We then apply the radiative transfer model of Corti and Peter (2009) to estimate the cloud radiative forcing caused by the detected cirrus clouds in Section 4. The results are compared to previous

[revised manuscript text omitted]

---

## Referee Report (RR1)

A second review of 'Climatological and radiative properties of mid-latitude cirrus clouds derived by automatic evaluation of lidar measurements'
By Kienast-Sjögren et al.
Submitted to atmospheric chemistry and physics

Generally, I'm satisfied with the authors' response to my previous comments. In their response to surface albedo, the authors used a value of 0.3 to demonstrate the global average IRF. It is OK in this paper, but the authors may notice that to obtain an accurate IRF locally, it is necessary to characterize surface albedo correctly.

My recommendation is acceptance.